# GAE: Unleashing Physical Potential of VLM with Generalizable Action Expert

**Mingyu Liu** [* 1 2 3]  **Zheng Huang** [* 1]  **Xiaoyi Lin** [1]  **Muzhi Zhu** [1]  **Canyu Zhao** [1]
**Yating Wang** [3]  **Haoyi Zhu** [3]  **Hao Chen** [† 1]  **Chunhua Shen** [† 1 4]

## Abstract

Vision-language models demonstrate strong reasoning and planning abilities, yet grounding these predictions into precise robot actions remains a central challenge. Existing Vision-Language-Action methods typically entangle reasoning and action generation, leading to limited generalization. We propose **G**eneralizable **A**ction **E**xpert (**GAE**), a task-agnostic model that converts sparse geometric plans into dense robot actions. Our approach introduces a sparse geometric interface: the VLM predicts sparse 3D waypoints representing high-level intention, while GAE maps these waypoints together with real-time point cloud observations to continuous action trajectories. GAE is pretrained on a large-scale point-cloud–trajectory dataset comprising **150k** trajectories from both simulation and real-world robots. To further improve efficiency and generalization, we introduce an **Action Pre-training, Point-cloud Fine-tuning (APPF)** scheme that decouples learning action dynamics from geometry grounding. After pretraining, GAE is frozen and reused across downstream tasks, requiring only lightweight fine-tuning of the VLM to produce the sparse interface. Experiments show that our method achieves strong performance and generalization across diverse visual domains, camera viewpoints, and natural language instructions.

## 1. Introduction

Vision-Language Models (VLMs) (Bai et al., 2025; Chen et al., 2024b; Wang et al., 2025b; Comanici et al., 2025) have demonstrated powerful capabilities in visual understanding, spatial reasoning, and task planning. However, translating these abilities into the physical world remains highly challenging. A prominent strategy to bridge this gap is the Vision-Language-Action (VLA) paradigm (Kim et al., 2024; Brohan et al., 2022; Zitkovich et al., 2023), which integrates reasoning and action within a unified architecture. Yet this approach exposes a critical paradox: achieving precise robotic control typically requires intensive fine-tuning on limited, domain-specific robotics datasets, which often induces catastrophic forgetting in the underlying VLM. Consequently, the model degrades its original knowledge and struggles to generalize to new tasks and environments.

While recent dual-system frameworks (Black et al.; Li et al., 2025; Huang et al., 2025; Li et al., 2024b) attempt to decouple "thinking" from "acting," the interface between these modules often remains poorly defined. Ideally, the VLM interprets scenes and instructions to generate high-level plans, while the action expert merely translates these plans into motion. In practice, however, the action module is still required to process semantically rich inputs, such as visual features (Bjorck et al., 2025; Li et al., 2024b) or abstract embeddings (Huang et al., 2025) from the VLM. This creates a fundamental conflict: the action policy must remain lightweight for real-time control, yet is simultaneously tasked with parsing complex high-level information. This semantic burden hinders cross-task training, limits generalization, and necessitates costly fine-tuning for new scenarios.

This naturally raises a key question: can we learn a generalizable action expert that operates with minimal or no semantic input (e.g., RGB images or VLM features), and instead relies purely on spatial geometry to produce precise robot actions? Achieving this requires a clean and structured interface between high-level reasoning and low-level control.

We propose to use a sparse 3D pose trajectory as this interface. The VLM predicts a small set of coarse keyframe end-effector poses that encode high-level intent, while the action expert maps these poses together with real-time uncolored point clouds to dense, executable action trajectories. Crucially, both the input guidance and the output of the action expert lie in the same action space, which frees the

[1]State Key Laboratory of CAD&CG, Zhejiang University [2]Shanghai Innovation Institute [3]Shanghai AI Laboratory [4]Ant Group. Correspondence to: Hao Chen <haochen.cad@zju.edu.cn>, Chunhua Shen <chunhua@me.com>.

*Proceedings of the 43rd International Conference on Machine Learning*, Seoul, South Korea. PMLR 306, 2026. Copyright 2026 by the author(s).

expert from interpreting task semantics and reduces its role to geometric motion refinement.

This separation enables a task-agnostic **G**eneralizable **A**ction **E**xpert (**GAE**) that can be trained once and reused across tasks without task-specific fine-tuning. The VLM only needs lightweight adaptation to emit sparse pose guidance, thereby avoiding catastrophic forgetting caused by dense trajectory supervision.

Training such a model is still non-trivial due to the scarcity of high-quality point cloud data and the high cost of jointly modeling geometry and actions. We address this with a large-scale pointcloud–trajectory dataset containing 150k trajectories from both simulation and real-world robots, and a two-stage **Action Pre-training, Pointcloud Fine-tuning (APPF)** scheme that first learns action dynamics from trajectories and then grounds them in geometry. This strategy substantially improves training efficiency and generalization. Extensive experiments demonstrate that our approach achieves strong performance and generalization across diverse visual domains, camera viewpoints, and natural language instructions.

In summary, our contributions are threefold:

- We formulate a sparse geometric interface for Vision-Language-Action systems and propose a task-agnostic **G**eneralizable **A**ction **E**xpert (**GAE**) that maps sparse 3D pose guidance and point cloud observations to dense action trajectories, effectively separating semantic reasoning from motor control.
- We curate a large-scale pointcloud–trajectory dataset containing **150k** trajectories from both simulation and real-world robots, and introduce an **Action Pre-training, Pointcloud Fine-tuning (APPF)** scheme that decouples learning action dynamics from geometry grounding, improving training efficiency and generalization.
- We demonstrate through extensive experiments that our approach achieves strong performance and generalization across diverse visual domains, camera viewpoints, and natural language instructions.

## 2. Related Works

### 2.1. VLM Spatial Reasoning

Equipping vision-language models (VLMs) with spatial reasoning capabilities has become an active research direction. Pre-trained on large-scale datasets, recent VLMs (Bai et al., 2025; Chen et al., 2024b; Comanici et al., 2025) demonstrate promising performance in understanding 3D spatial relationships and achieve competitive results on spatial reasoning benchmarks (Zhang et al., 2021; Azuma et al., 2022; Ma et al., 2022). Numerous studies (Wu et al., 2025a; Cai et al., 2024; Zhu et al., 2024; Chen et al., 2024a; Yuan et al.,

2024; Song et al., 2025; Yang et al., 2025) further enhance spatial reasoning by either fine-tuning on large-scale 3D datasets or explicitly incorporating 3D information. In contrast to these specialized designs, we adopt a widely used general-purpose VLM as our backbone, rather than models specifically trained for 3D data, to evaluate the generality of our framework. Our paradigm is model-agnostic and readily compatible with a broad range of existing VLMs.

### 2.2. Vision-Language-Action Models

Conventional VLA models (Brohan et al., 2022; Zitkovich et al., 2023; Cheang et al., 2024; Ha et al., 2023; Kim et al., 2024; Black et al.; Wen et al., 2025b; Team et al., 2024) typically adopt monolithic architectures that map vision and language directly to actions. Recent dual-system frameworks introduce hierarchical designs with a high-level planner and a low-level action expert, communicating through intermediate representations such as trajectories (Li et al., 2024b; de Bakker et al., 2025; Huang et al., 2025), visual features (Li et al., 2025; Bjorck et al., 2025), or attention-based signals (Black et al.; AgiBot-World-Contributors et al., 2025). While this structure provides partial decoupling, the choice of intermediate representation remains challenging. Trajectory-following approaches like Hamster (Li et al., 2024b) still require the low-level policy to process complex structured information, increasing the burden on the action expert. Moreover, advanced works (Qi et al., 2025) do not fully explore the potential of a learnable action policy, leaving the design of effective intermediate representations and their tolerance to errors largely open.

### 2.3. Generalizable Action Expert

Learning generalizable action policies is a longstanding goal in robotics. Action experts, such as diffusion-based or autoregressive policies (Chi et al., 2023; Zhao et al., 2023), are computationally efficient but have limited capacity, making large-scale multi-task training difficult. A common strategy is to provide intermediate guidance signals, including object poses (Deng et al., 2020), keypoints (Manuelli et al., 2019), affordances (Wu et al., 2025b), and semantically segmented point clouds (Zhu et al., 2023), which offer coarse action structure and shift the policy's focus to low-level refinement. While such methods improve over unguided policies, their guidance is often task-specific or semantically rich, limiting transfer and requiring additional adaptation. In contrast, our method uses sparse 3D end-effector pose trajectories as minimal geometric guidance, refined into dense actions using real-time point cloud observations.

## 3. Method

Figure 1 illustrates the overall pipeline of our method, which employs 3D pose trajectories as an interface between the

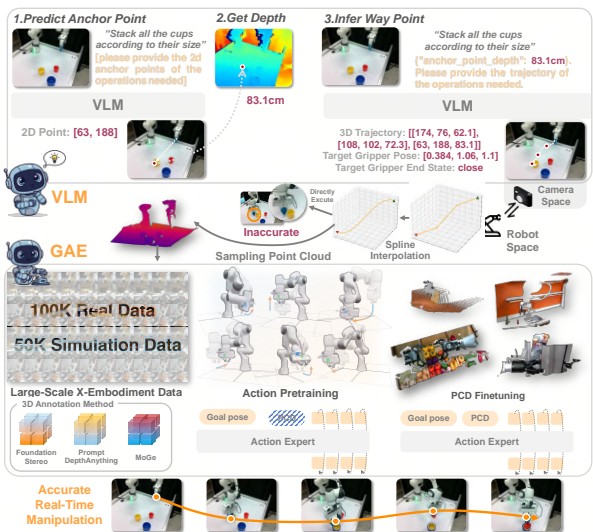

*Figure 1.* **The pipeline of our proposed method.** Our approach begins with a VLM predicting a sparse set of 3D waypoints directly in the camera frame, preserving its vision-centric knowledge. These sparse points are then transformed and interpolated via a B-spline into a continuous and smooth pose trajectory to provide dense guidance for a low-level action expert.

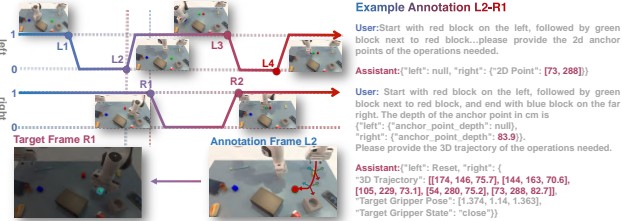

*Figure 2.* **Overview of our data annotation pipeline.** We construct our SFT dataset by first selecting keyframes based on gripper state changes. $L_n$ means the $n$th keyframe of left hand, and $R_n$ means the $n$th keyframe of right hand. For Single Arm dataset, the annotation pipeline is same.

high-level VLM and the low-level action expert. The VLM reasons over 2D keypoints together with their depth values, and outputs a sparse set of 3D waypoints as well as a final end-effector pose associated with the target keypoint. We adopt Qwen2.5VL (Bai et al., 2025) as the VLM backbone; comparisons with alternative backbones and model scales are provided in Appendix F.4.

The predicted waypoints and target pose are transformed from the camera frame to the robot base frame, followed by B-spline interpolation to obtain a smooth continuous trajectory. We sample one guidance pose every 8 steps along this trajectory and feed these poses to the Action Expert. Additional implementation details are provided in Sections 3.1 and 3.4.

Most existing Vision-Language-Action (VLA) models predict action coordinates directly in the robot base frame. While effective in controlled settings, this formulation places the burden of learning camera-to-robot transformations largely on the VLM, often with limited explicit geometric supervision. As a consequence, the model may rely on dataset-specific spatial correlations rather than developing robust, transferable spatial representations, which can negatively impact generalization. In contrast, our approach predicts waypoints in the camera frame, which aligns naturally with the VLM's image-centric pre-training. This design simplifies the spatial reasoning problem faced by the VLM and allows it to focus on interpreting visual observations and instructions, while leaving geometric grounding and motion refinement to the action expert.

## 3.1. Lightweight Finetuning of VLM

Recent studies show that vision-language models pretrained on internet-scale data exhibit strong performance in 2D localization and coarse spatial reasoning. However, directly inferring target robot poses from images and instructions remains non-trivial. To enable this capability while preserving the VLM's general-purpose knowledge, we lightly fine-tune the VLM using a small amount of carefully constructed supervision.

Specifically, we identify keyframes at moments when the gripper's kinematic state changes (e.g., opening or closing), as these often correspond to meaningful interaction events. Based on these keyframes, we build a supervised fine-tuning (SFT) dataset that provides sparse pose-level supervision. The overall annotation pipeline is illustrated in Figure 2. This design supplies explicit spatial grounding signals without requiring dense trajectory-level supervision, thereby limiting the extent of task-specific adaptation.

For each selected keyframe, the VLM is first prompted to predict a 2D anchor point corresponding to the target grasping or placing location. Using depth information, we recover the associated 3D coordinate $(u, v, d)$. The VLM then predicts a sparse set of intermediate waypoints along the motion path, together with a final target end-effector pose. These waypoints and the target pose are subsequently interpolated using a spline to form a continuous end-effector pose trajectory, which serves as guidance for the action expert.

As shown in Table 5, after this lightweight fine-tuning, the VLM retains strong language understanding and general reasoning capabilities, indicating that our supervision preserves the model's general-purpose knowledge.

## 3.2. Closed-Loop Trajectory Update

To handle external disturbances during execution, we make our VLM performs asynchronous secondary inference while executing the current trajectory. When updated waypoints

from the VLM become available, we merge the new plan with the executing trajectory at a future time step.

Let $\boldsymbol{T}_0(t_2)$ denote the predicted robot pose at the merge time and $\mathcal{P}' = \{\boldsymbol{p}_i'\}_{i=1}^{n'}$ the new waypoint sequence. We first select the closest waypoint

$$k^* = \arg \min_{1 \leq i \leq n'} \left\| \boldsymbol{p}_i' - \boldsymbol{T}_0(t_2) \right\|_2. \tag{1}$$

To avoid backtracking, we require directional consistency with the local path direction $\hat{\boldsymbol{v}}_{\text{path}}$:

$$\gamma = (\boldsymbol{p}_{k^*}' - \boldsymbol{T}_0(t_2))^\top \hat{\boldsymbol{v}}_{\text{path}}, \qquad \gamma > 0. \tag{2}$$

All earlier waypoints are discarded, and a transition spline connects $(\boldsymbol{T}_0(t_2), \dot{\boldsymbol{T}}_0(t_2))$ to the remaining trajectory to ensure smooth position and velocity continuity.

This closed-loop merging enables online correction of execution deviations without stopping motion.

### 3.3. Directly Using IK for Manipulation

Given that our VLM predicts sparse 3D end-effector waypoints, a natural question is whether these waypoints can be directly converted into joint trajectories using inverse kinematics (IK), without learning an additional action expert. Indeed, for simple tasks in uncluttered environments, we find that applying IK to the interpolated end-effector trajectory can already produce reasonable motions.

However, relying solely on IK presents several practical limitations. First, IK-based execution does not account for environmental geometry beyond the target pose, and therefore cannot adapt trajectories to avoid obstacles or reason about contact-rich interactions. This becomes particularly problematic for tasks requiring fine-grained manipulation or complex motion patterns, such as inserting objects into tight spaces, stacking multiple blocks. Second, IK is highly sensitive to errors in the predicted end-effector poses. When the depth estimation or waypoint prediction is noisy, even small geometric inaccuracies can lead to infeasible joint solutions or unstable motions.

We empirically evaluate direct IK-based execution in both simulation Tab 2 and real-world experiments Tab 4, and observe a substantial performance gap compared to our full system. These results indicate that sparse pose guidance alone is insufficient for robust manipulation.

Motivated by these observations, we introduce a geometry-aware action expert that refines the sparse end-effector trajectory using real-time point cloud observations. This learned refinement enables the system to adjust motions based on local geometry, improve robustness to pose noise, and handle contact-rich and cluttered scenarios, while still benefiting from the high-level guidance provided by the VLM.

### 3.4. Training Generalizable Action Expert

Developing a generalizable action expert presents two main challenges: obtaining high-quality point cloud data at scale, and efficiently training a model that conditions on both guidance poses and point clouds. We describe our solutions to these challenges respectively.

**Data Preparation.** Training a generalizable action expert requires large-scale high-quality point cloud annotations from both simulation and real-world environments. In simulation (Mu et al., 2025; Mees et al., 2022; Liu et al., 2023; James et al., 2019), accurate depth and camera parameters allow us to directly replay trajectories and obtain reliable point clouds.

Real-world datasets, however, often lack dense and accurate depth annotations, which limits their usefulness for learning geometry-conditioned policies. To address this issue, we re-annotate several large-scale real-world robot datasets (Khazatsky et al., 2025; AgiBot-World-Contributors et al., 2025; Walke et al., 2023) with improved depth and point cloud estimates, enabling the construction of a unified pointcloud–trajectory dataset spanning both simulation and real-world settings.

We further standardize all point clouds through foreground cropping and uniform downsampling, and use exclusively uncolored point clouds to encourage the action expert to focus on geometric structure rather than visual semantics. Detailed dataset sources, annotation pipelines, and statistics are provided in Appendix A.

**Action Pre-training, Pointcloud Fine-tuning (APPF).** Training our generalized action expert with extensive point cloud and trajectory data concurrently poses a significant efficiency challenge. We identified that the expert's role can be decomposed into two core skills: basic trajectory-following and environment-aware trajectory refinement using point clouds. To avoid the high cost and suboptimal learning that can result from training these coupled skills together, we introduce the "Action Pre-training, Pointcloud Fine-tuning" (**APPF**) paradigm. We first pre-train the expert on large batches of pure trajectory data (up to a batch size of 31,824) to master motion following, and then fine-tune it with point cloud data to learn refinement. Our experiments confirm this decoupled approach achieves faster convergence and substantially improves data utilization efficiency. Now we detail the architecture of our proposed model and its training process:

**Model Architecture.** Our action expert's architecture is inspired by the design of 3D Diffusion Policy (Ze et al., 2024). It integrates multimodal sensory inputs to guide the action generation process. The full set of conditioning inputs $\mathcal{C}_A$ for the action expert at any given timestep $t$ is

defined as:
$$\mathcal{C}_A = \{S_t, \mathcal{P}_g, O_{pcd}\} \tag{3}$$
where $S_t$ represents the robot's proprioceptive state, including information like joint positions and gripper state. $\mathcal{P}_g$ is the guidance pose, which is sampled from the continuous trajectory $\mathcal{T}(t)$ generated by the high-level VLM. $O_{pcd}$ is the cropped point cloud observation of the local environment, which is **uncolored**.

Each component of $\mathcal{C}_A$ is processed by a dedicated encoder (MLPs for $S_t$ and $\mathcal{P}_g$, and a PointNet-based encoder for $O_{pcd}$) to produce a final conditioning feature vector $f_t^A$.
$$f_t^A = \text{concat}(f_t^s, f_t^g, f_t^{pc}) \tag{4}$$

**Conditional Diffusion Model Training.** We model the action expert as a conditional diffusion policy. Instead of directly predicting an action, the policy learns to reverse a Gaussian diffusion process, iteratively refining a noisy action into a clean one, conditioned on $f_t^A$. The policy is parameterized as a noise prediction network $\epsilon_\theta$.

During training, we sample a ground-truth action $a_t^0$ from the expert demonstration dataset $\mathcal{D}$. We then create a noisy action $a_t^k$ by adding $k$ steps of Gaussian noise according to the noise schedule $\bar{\alpha}_k$:
$$a_t^k = \sqrt{\bar{\alpha}_k}a_t^0 + \sqrt{1 - \bar{\alpha}_k}\epsilon \tag{5}$$
where $\epsilon \sim \mathcal{N}(0, \mathbf{I})$ is random Gaussian noise. The noise prediction network $\epsilon_\theta$ is trained to predict the added noise $\epsilon$ based on the noisy action $a_t^k$, the diffusion step $k$, and the conditioning feature $f_t^A$. The learning objective is to minimize the L2 error on the predicted noise:
$$\mathcal{L}_{AE} = \mathbb{E}_{k \sim [1,K], a_t^0 \sim \mathcal{D}, \epsilon \sim \mathcal{N}(0,\mathbf{I})} \left[ \|\epsilon - \epsilon_\theta(a_t^k, k, f_t^A)\|^2 \right] \tag{6}$$

At inference time, an action is generated by starting with a random noise vector $a_t^K \sim \mathcal{N}(0, \mathbf{I})$ and iteratively applying the learned network $\epsilon_\theta$ to denoise it over $K$ steps, finally yielding a clean action $a_t^0$. In our experiments, the action chunk is set to 8, the action space is end-effector space.

This formulation fits our "Action Pre-training, Point Cloud Fine-tuning" paradigm. During pre-training, the point cloud feature within $f_t^A$ is masked out, training the diffusion model to follow trajectories. During fine-tuning, the full conditioning vector is used, enabling the model to refine its actions based on environmental context. More training details can be found in Appendix D.

## 4. Simulation Experiments

**Setup**. Our experimental validation in simulation is conducted on both single arm simulation environment SIMPLER (Li et al., 2024a) and dual arm simulation environment RoboTwin (Chen et al., 2025). We analyze the effect

of varying the number of SFT steps in Section 4.1, covering settings from zero-shot manner to more extensive fine-tuning.

To further demonstrate that our method mitigates catastrophic forgetting and preserves the generalization ability of the VLM, we evaluate cross-environment transfer of single-arm skills. Specifically, a VLM fine-tuned on SIMPLER is directly deployed on ManiSkill without any additional adaptation. The results show that the transferred model achieves performance comparable to the expert policy, indicating that our lightweight SFT does not overfit to the source environment and maintains strong generalization.

### 4.1. Results

**SIMPLER** Many tasks in the SIMPLER benchmark involve cluttered scenes and require non-trivial obstacle avoidance, where pure inverse-kinematics (IK) based execution often fails due to collisions or kinematic infeasibility. In such cases, successful execution relies not only on accurate end-effector pose prediction, but also on generating collision-aware and dynamically feasible motion trajectories. Results are shown in Table 1.

**RoboTwin** We present a comprehensive comparison of our model against existing generalist and expert models across 11 tasks in the RoboTwin benchmark, categorized into short, middle, and long horizons in Table 2. Our model surpasses popular generalist models on all tasks. For short-horizon tasks, we achieve performance on par with the DP3 expert model. Our primary advantage is demonstrated in long-horizon tasks that require VLM-based planning. On these tasks, where specialized expert models almost universally fail, our model shows exceptional capability, achieving a 60% average success rate. Notably, while other generalist models like $\pi_0$ and RDT require task-specific fine-tuning after multi-task training to achieve a moderate performance, our method does not. The fine-tuning steps for our VLM are detailed in Table 3. We present more generalization experiments of RoboTwin in Appendix B.3.

**Dynamic Failure Recovery.** Effective manipulation also requires recovery from physical execution failures (e.g., object slippage or grasp misses). Figure 3 shows a qualitative example of our method's closed-loop correction behavior. During a "Stack Cube" task, the target object slipped from the gripper due to low friction. The system immediately truncated the original plan and generated a corrective "re-grasp" sequence. This behavior was not explicitly hard-coded but emerged from the combination of high-frequency visual feedback and our receding horizon control strategy, effectively countering the critique that spline-based detokenization lacks interaction robustness.

*Table 1.* **SimplerEnv evaluation across different models on Google Robot tasks.** Drawer: Open/Close Drawer.

| Model | Visual Matching | | | | Variant Aggregation | | | | Overall |
|---|---|---|---|---|---|---|---|---|---|
| | Pick Coke | Move Near | Drawer | Avg. | Pick Coke | Move Near | Drawer | Avg. | Avg. |
| RT-1-X (Brohan et al., 2022) | 56.7% | 31.7% | 59.7% | 53.4% | 49.0% | 32.3% | 29.4% | 39.7% | 46.6% |
| Octo-Base (Team et al., 2024) | 17.0% | 4.2% | 22.7% | 16.8% | 0.6% | 3.1% | 1.1% | 1.2% | 9.0% |
| $\pi_0$ (Black et al.) | 72.7% | 65.3% | 38.3% | 58.8% | 75.2% | 63.7% | 25.6% | 54.8% | 56.8% |
| $\pi_0$-FAST (Pertsch et al., 2025) | 75.3% | 67.5% | 42.9% | 61.9% | 77.6% | 68.2% | **31.3%** | 59.0% | 60.5% |
| OpenVLA (Kim et al., 2024) | 16.3% | 46.2% | 35.6% | 32.7% | 54.5% | 47.7% | 17.7% | 39.8% | 33.8% |
| GR00T-N1 (Bjorck et al., 2025) | 47.0% | 70.0% | 18.1% | 45.0% | 78.8% | 62.5% | 13.2% | 51.5% | 48.4% |
| SoFAR (Qi et al., 2025) | 92.3% | 91.7% | 40.3% | 74.8% | 90.7% | 74.0% | 29.7% | 64.8% | 69.8% |
| **Ours** | **94.0%** | **92.5%** | **62.5%** | **83.0%** | **91.9%** | **75.0%** | 28.3% | **65.1%** | **74.1%** |

*Table 2.* Performance comparison between our multi-task generalist model and various single-task expert models across short, middle, and long-horizon tasks. The best performance in each row is **bolded**, and the second-best is underlined.

| Category | Task Name | Generalist Model | | | | | Expert Model | | |
|---|---|---|---|---|---|---|---|---|---|
| | | Ours | $\pi_0$ | $\pi_0$-FAST | $\pi_{0.5}$ | RDT | ACT | DP | DP3 |
| **Short Horizon** | Click Bell | **0.93** | 0.44 | 0.85 | 0.06 | 0.80 | 0.58 | 0.54 | 0.90 |
| | Grab Roller | 0.99 | 0.96 | 0.97 | **1.00** | 0.74 | 0.94 | 0.98 | 0.98 |
| | Lift Pot | 0.95 | 0.72 | 0.76 | 0.00 | 0.84 | 0.88 | 0.39 | **0.97** |
| | Place Phone Stand | **0.54** | 0.35 | 0.38 | **0.54** | 0.15 | 0.02 | 0.13 | 0.44 |
| | Avg. | **0.82** | 0.62 | 0.74 | 0.40 | 0.63 | 0.61 | 0.51 | **0.82** |
| **Middle Horizon** | Handover Mic | **1.00** | 0.98 | 0.87 | 0.37 | 0.90 | 0.85 | 0.53 | **1.00** |
| | Place A2B Left | **0.64** | 0.31 | 0.36 | **0.64** | 0.03 | 0.01 | 0.02 | 0.46 |
| | Place Bread Basket | **0.66** | 0.17 | 0.22 | 0.60 | 0.10 | 0.06 | 0.14 | 0.26 |
| | Stack Block Two | **0.88** | 0.42 | 0.46 | 0.85 | 0.21 | 0.25 | 0.07 | 0.24 |
| | Avg. | **0.77** | 0.47 | 0.48 | 0.62 | 0.31 | 0.29 | 0.19 | 0.49 |
| **Long Horizon** | Blocks Ranking RGB | **0.78** | 0.19 | 0.23 | 0.63 | 0.03 | 0.01 | 0.00 | 0.03 |
| | Blocks Ranking Size | **0.53** | 0.07 | 0.10 | 0.33 | 0.00 | 0.00 | 0.01 | 0.02 |
| | Stack Block Three | 0.54 | 0.17 | 0.20 | **0.55** | 0.02 | 0.00 | 0.00 | 0.01 |
| | Avg. | **0.60** | 0.14 | 0.18 | 0.50 | 0.02 | 0.003 | 0.003 | 0.02 |

*Table 3.* Training Steps for Different Methods (GPU num × batch-size × steps).

| Method | Ours | ACT | DP | DP3 | $\pi_0$ | $\pi_0$-FAST | $\pi_{0.5}$ | RDT |
|---|---|---|---|---|---|---|---|---|
| **Training Steps (All tasks)** | 8*32*1k | 0 | 0 | 0 | 8*32*1k | 8*32*1k | 8*32*5k | 8*32*1k |
| **Single task fine-tuning** | 0 | 32*10k | 32*10k | 32*30k | 8*32*5k | 8*32*5k | 0 | 8*32*5k |

## 5. Real-World Experiments

### 5.1. Setup

**In-Distribution (ID) Setup.** For each task, we collect 20 demonstration trajectories in the training workspace. To systematically control the initial object placement, we discretize the workspace into 4 predefined ID positions, and record 5 trajectories per position. We use a localization light marker to indicate and log the exact position used in each rollout, ensuring consistent and repeatable placement across data collection.

**Out-of-Distribution (OOD) Setup.** To evaluate position generalization, we additionally select 4 extra object positions that never appear in the training set. For each task, we test the trained policy 5 times at each unseen position. The same localization light marker is used to record each OOD position, enabling a controlled comparison between ID and OOD settings. The detail of how we design ID and OOD position is presented in Appendix C

**Platform.** Our real-world setup includes both single-arm and dual-arm platforms, aligned with our simulation environments. For single-arm experiments, we use a Franka

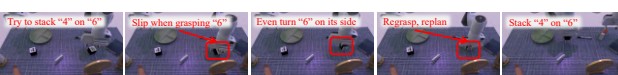

*Figure 3.* **Closed-Loop Failure Recovery.** Our method recovers from execution failures via closed-loop replanning. (1) **Slip:** The initial grasp on the block fails. (2) **Re-planning:** Asynchronous secondary inference detects the gripper-object mismatch and updated object pose. (3) **Correction:** A re-grasp trajectory is generated and merged online, allowing the task to complete successfully.

Research 3 robot equipped with a UMI gripper (Chi et al., 2024). For dual-arm experiments, we use two aligex piperX to assemble a dual-arm robot. In both setups, a third-person RealSense D435 camera is mounted in a fixed position to capture environmental observations.

**Protocols.** For the real-world evaluation, 20 trials are performed per method for each task. All methods are evaluated under closely matched randomized initial scene configurations for each trial. Expert-based baselines are fine-tuned separately for each individual task, while generalist methods are co-trained on all tasks. Our result is presented in Table 4. Experiment setups and task definition are briefly shown in Figure 4. Detailed training and evaluation settings of our method can be found in Appendix C.

## 6. Ablation Studies

**Ablation on Training Steps.** We study the effect of different Vision-Language Model (VLM) fine-tuning budgets, from zero-shot to heavy fine-tuning that degrades language capability, in Table 5. On the RoboTwin tasks, we evaluate success rates together with VLM's language performance. Results show that using the generalizable action expert allows performance to saturate with substantially fewer supervised fine-tuning (SFT) steps (Figure 5), reducing the required VLM adaptation and better preserving language generalization.

*Table 5.* **SFT step ablation study.** VLM's language ability degrades as sft steps increase

| SFT Steps | 0 | 500 | 1k | 1.5k | 2k | 2.5k | 3k | 3.5k | 4k |
|---|---|---|---|---|---|---|---|---|---|
| **VLM + IK** | 0.04 | 0.26 | 0.32 | 0.34 | 0.40 | 0.43 | 0.47 | 0.52 | 0.52 |
| **VLM + GAE** | 0.10 | 0.44 | 0.56 | 0.56 | 0.57 | 0.57 | 0.58 | 0.58 | 0.58 |
| **MMLU (Hendrycks et al., 2020)** | 70.1 | 61.3 | 49.3 | 41.8 | 37.3 | 30.0 | 29.9 | 29.9 | 29.4 |

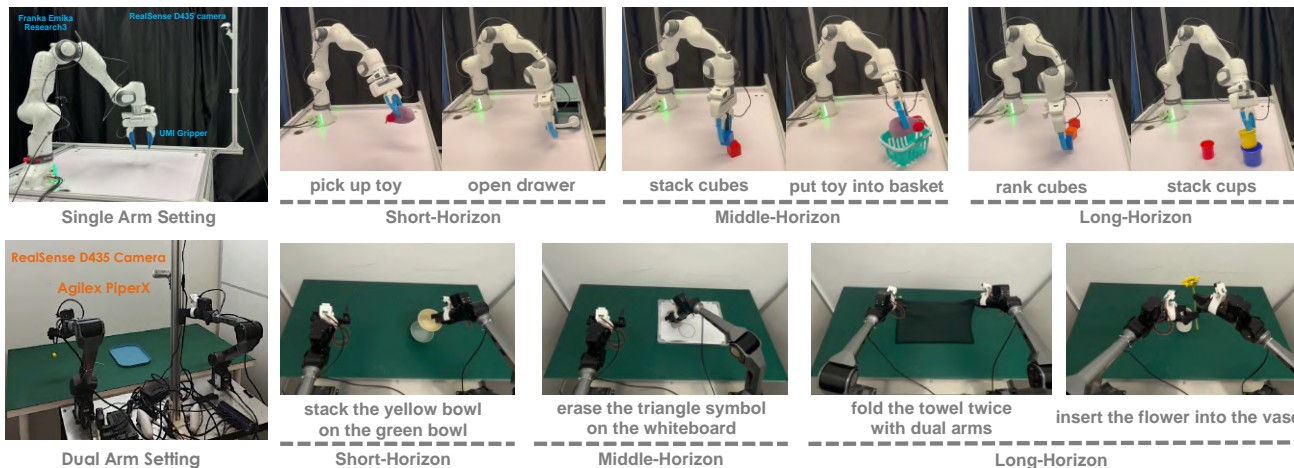

*Figure 4.* **Real-World Task Setting.** We evaluate our method on both single-arm and dual-arm platforms, and design tasks spanning different horizons. We include fine-grained dual-arm manipulation tasks, such as towel folding and flower insertion, to further assess the model's capability for precise and coordinated manipulation.

*Table 4.* **Real world performance of our method and baselines.** We report success rate drops under OOD settings.

| Method | Short Horizon | | | Middle Horizon | | | Long Horizon | | | |
|---|---|---|---|---|---|---|---|---|---|---|
| | pick toy | open drawer | stack bowl | stack cubes | toy basket | erase symbol | rank cubes | stack cups | fold towel | insert flower |
| ACT (Zhao et al., 2023) | 0.65→0.35 | 0.55→0.35 | 0.45→0.25 | 0.40→0.25 | 0.50→0.30 | 0.35→0.20 | 0.10→0.05 | 0.00→0.00 | 0.05→0.05 | 0.00→0.00 |
| DP (Chi et al., 2023) | 0.85→0.60 | 0.75→0.50 | 0.55→0.40 | 0.40→0.25 | 0.45→0.30 | 0.40→0.25 | 0.15→0.10 | 0.00→0.00 | 0.10→0.05 | 0.05→0.00 |
| DP3 (Ze et al., 2024) | 0.90→0.70 | 1.00→0.45 | 0.65→0.45 | 0.60→0.45 | 0.75→0.55 | 0.55→0.40 | 0.20→0.10 | 0.15→0.10 | 0.20→0.10 | 0.15→0.00 |
| OpenVLA (Kim et al., 2024) | 0.85→0.60 | 0.40→0.25 | 0.30→0.20 | 0.35→0.20 | 0.55→0.35 | 0.25→0.15 | 0.45→0.25 | 0.20→0.10 | 0.15→0.10 | 0.10→0.05 |
| $\pi_0$ (Black et al.) | 0.75→0.55 | 0.55→0.40 | 0.40→0.30 | 0.45→0.30 | 0.60→0.40 | 0.35→0.25 | 0.50→0.30 | 0.10→0.05 | 0.20→0.10 | 0.15→0.10 |
| $\pi_0$-FAST (Pertsch et al., 2025) | 0.80→0.60 | 0.60→0.45 | 0.45→0.30 | 0.50→0.35 | 0.65→0.45 | 0.40→0.30 | 0.55→0.35 | 0.15→0.10 | 0.25→0.15 | 0.20→0.10 |
| $\pi_{0.5}$ (Intelligence et al.) | 1.00→**0.95** | 0.85→0.50 | 0.80→0.30 | 0.50→0.30 | 0.60→0.40 | 0.40→0.25 | 0.50→0.30 | 0.45→0.15 | 0.20→0.15 | 0.20→0.10 |
| SoFAR (Qi et al., 2025) | 0.80→0.65 | 0.70→0.55 | 0.60→0.45 | 0.60→0.45 | 0.70→0.55 | 0.55→0.40 | 0.60→0.45 | 0.35→0.25 | 0.50→0.40 | 0.45→0.35 |
| HAMSTER (Li et al., 2024b) | 0.70→0.50 | 0.55→0.40 | 0.45→0.30 | 0.45→0.30 | 0.55→0.40 | 0.40→0.30 | 0.45→0.30 | 0.30→0.20 | 0.40→0.30 | 0.35→0.05 |
| GR00T-N1 (Bjorck et al., 2025) | 0.70→0.50 | 0.60→0.45 | 0.50→0.30 | 0.55→0.40 | 0.60→0.45 | 0.45→0.30 | 0.50→0.30 | 0.25→0.15 | 0.35→0.25 | 0.30→0.20 |
| VLM+IK | 0.75→0.60 | 0.60→0.45 | 0.45→0.30 | 0.40→0.25 | 0.50→0.35 | 0.35→0.25 | 0.50→0.30 | 0.30→0.20 | 0.25→0.15 | 0.20→0.10 |
| VLM+GAE (Ours) | 0.90→0.90 | 0.80→**0.70** | 0.75→**0.65** | 0.75→**0.65** | 0.80→**0.70** | 0.70→**0.60** | 0.70→**0.60** | 0.55→**0.45** | 0.65→**0.55** | 0.55→**0.45** |

**Robustness to Input Depth Noise.** In real-world settings, accurate ground-truth depth is often unavailable and depth estimates can be noisy. To evaluate robustness under such conditions, we inject Gaussian noise with varying maximum amplitudes into the input depth and report success rates in Table 6. Notably, the same noisy depth is used both for generating pose guidance and for constructing the point clouds consumed by the GAE. As the noise level increases, direct IK execution degrades rapidly and fails on many tasks. In contrast, using GAE leads to a much slower performance degradation across all noise scales.

*Table 6.* **Robustness analysis against depth estimation noise.** GAE is more robust to depth noise than IK

| Task Name | Noise Level ($\sigma_{max}$) | | | | | | | |
|---|---|---|---|---|---|---|---|---|
| | 0 cm | | 1.0 cm | | 10.0 cm | | 50.0 cm | |
| | IK | GAE | IK | GAE | IK | GAE | IK | GAE |
| Place a2b left | 0.40 | 0.55 | 0.35 | 0.50 | 0.20 | 0.35 | 0.10 | 0.25 |
| Place a2b right | 0.30 | 0.45 | 0.25 | 0.40 | 0.15 | 0.30 | 0.05 | 0.10 |
| Place object scale | 0.65 | 0.75 | 0.25 | 0.55 | 0.25 | 0.45 | 0.05 | 0.35 |
| Stack blocks three | 0.35 | 0.55 | 0.30 | 0.50 | 0.00 | 0.30 | 0.00 | 0.10 |
| Stack bowl three | 0.65 | 0.75 | 0.40 | 0.60 | 0.10 | 0.35 | 0.00 | 0.30 |
| Blocks ranking size | 0.20 | 0.40 | 0.20 | 0.35 | 0.00 | 0.25 | 0.00 | 0.20 |
| Blocks ranking rgb | 0.40 | 0.55 | 0.35 | 0.50 | 0.05 | 0.30 | 0.10 | 0.15 |
| Average | 0.42 | **0.57** | 0.30 | **0.49** | 0.11 | **0.33** | 0.04 | **0.21** |

**Ablation on Guidance Pose's Noise.** We evaluate **GAE**'s robustness to goal-pose perturbations in Table 7, and observe optimal performance on the tasks in Table 2 when training with a noise scale of 0.1. This noise is injected during training of GAE to mimic variability in VLM-generated trajectories, improving generalization. No additional noise is applied at inference time. Resuls show proper noise magnitude is critical: insufficient noise causes overfitting, while excessive noise degrades performance.

*Table 7.* **Noise scale ablation.** Introducing a suitable noise shift during training enhances performance on downstream tasks.

| Noise scale | Short Horizon | Middle Horizon | Long Horizon |
|---|---|---|---|
| 0.00 | 0.71 | 0.67 | 0.43 |
| 0.05 | 0.76 | 0.74 | 0.49 |
| 0.10 | **0.82** | **0.77** | **0.60** |
| 0.20 | 0.80 | 0.72 | 0.59 |
| 0.50 | 0.68 | 0.68 | 0.41 |

**Different Interpolation Choices.** We compare B-spline interpolation with simple linear interpolation. B-splines are used by default as they produce smoother trajectories that

better match typical robotic motion. Results show that linear interpolation can achieve reasonably good performance.

*Table 8.* **Ablation study of different interpolation methods.**

|  | Short Horizon | Middle Horizon | Long Horizon |
|---|---|---|---|
| Linear | 0.75 | 0.69 | 0.58 |
| B-spline | **0.82** | **0.77** | **0.60** |

**Different Training Strategy.** To evaluate the proposed "Action Pre-training, Point Cloud Fine-tuning" (**APPF**) paradigm, we compare it with a conventional end-to-end strategy that jointly trains on trajectories and point clouds from the beginning. Results show that **APPF** consistently outperforms joint training. By pre-training only on trajectory data, **APPF** avoids the computational overhead of point cloud processing, allowing substantially larger batch sizes (up to 32,768) and faster convergence (Figure 6). We further report a quantitative comparison of the two strategies at 50k training steps in Table 9.

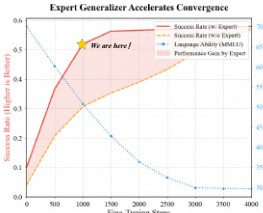

*Figure 5.* Training steps.          *Figure 6.* Training strategy.

*Table 9.* **Ablation study of different training strategy.** With our APPF strategy, the model outperforms the vanilla setting in all horizon tasks.

|  | Short Horizon | Middle Horizon | Long Horizon |
|---|---|---|---|
| w/o APPF | 0.72 | 0.66 | 0.50 |
| APPF | **0.82** | **0.77** | **0.60** |

**Ablation on Keyframe Extraction.** Our kinetic-keyframe rule is one specific instantiation of a more general principle, rather than a claim that sparse supervision must rely on gripper kinematics. To verify that the overall framework does not critically depend on this hand-designed heuristic, we compare it against a simple *uniform* keyframe sampling baseline under a matched supervision budget on RoboTwin. As shown in Table 10, uniform sampling remains highly competitive across all horizons, while the kinetic variant gives a small but consistent improvement on most settings.

*Table 10.* **Ablation on keyframe extraction strategy.**

|  | Short Horizon | Middle Horizon | Long Horizon |
|---|---|---|---|
| Uniform | 0.81 | 0.74 | **0.61** |
| Kinetic (ours) | **0.82** | **0.77** | 0.60 |

**Ablation on VLM Finetuning Setup.** Our default setup uses *multi-benchmark joint finetuning* for the simulation is jointly finetuned across all simulation benchmarks. To clarify the fairness of this protocol, we additionally finetune the VLM in a *single-benchmark specialized* manner on RoboTwin only, which matches the setting used by the baselines in our paper. As shown in Table 11, single-benchmark finetuning yields a modest improvement on its own test benchmark, as expected. Crucially, however, our default multi-benchmark setting, which is strictly harder, still surpasses conventional baselines trained under the easier single-benchmark protocol, while also requiring fewer finetuning steps.

*Table 11.* **Ablation on VLM finetuning setup.**

|  | Short Horizon | Middle Horizon | Long Horizon |
|---|---|---|---|
| Multi-bench (ours) | 0.82 | 0.77 | **0.60** |
| Single-bench | **0.85** | **0.78** | 0.58 |

## 7. Limitations and Future Work

Our current interface between the VLM and the action expert uses sparse 3D waypoints, which provide an effective and simple abstraction but are manually designed. A promising direction is to explore more expressive and compact continuous representations that enable end-to-end optimization. In addition, our method relies on external depth sensing to construct point clouds, which may introduce estimation error and additional computation. Although our model show strong robustness to moderate depth noise, tighter integration of geometry estimation and action generation could further improve efficiency and reliability. Finally, real-world deployment currently needs an initial camera extrinsic calibration, advances in camera pose estimation may further simplify deployment and broaden real-world applicability.

## 8. Conclusion

We present GAE, improving generalization in Vision-Language-Action models by cleanly separating high-level reasoning from low-level control through a sparse 3D pose interface. Central to our approach is a GAE, trained with a curated point cloud dataset and a two-stage Action Pre-training, Pointcloud Fine-tuning paradigm. Experiments demonstrate robust performance and strong generalization across diverse tasks, visual domains, and language instructions, without task-specific fine-tuning. Our work represents a meaningful step toward scalable robotic manipulation.

## Acknowledgement

This work was supported by the National Natural Science Foundation of China (No. 62576315)

## Impact Statement

This work aims to advance the robotics community's understanding of how high-level semantic reasoning can be systematically decoupled from low-level action execution. We investigate effective strategies for introducing a sparse geometric interface between vision-language models and action policies to enable reusable and generalizable manipulation skills. We posit that a compact representation in the form of sparse 3D pose trajectories is sufficient to convey task intent for a wide range of manipulation behaviors, while allowing a geometry-aware action expert to handle local motion refinement. From the perspective of the broader machine learning community, this work provides a principled framework for modularizing foundation models and efficiently adapting them to embodied decision-making problems.

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

## A. Detail Annotation Pipeline and Description of Pointcloud Dataset

Training a well-generalized action expert model requires a large volume of high-quality 3D point cloud annotations from both real-world and simulated environments. In simulation, obtaining high-fidelity point clouds is straightforward due to precise depth and camera data. We replayed trajectories from multiple simulators, including RoboTwin (Mu et al., 2025; Chen et al., 2025), CALVIN (Mees et al., 2022), LIBERO (Liu et al., 2023) and RLBench (James et al., 2019), yielding a total of 50k trajectories accompanied by high-accuracy point clouds and joint poses.

In contrast, real-world datasets present significant challenges. Most existing robot manipulation datasets lack high-quality depth annotations and accurate camera calibration parameters. Their depth information typically comes from depth sensors, resulting in sparse and incomplete depth maps. This severely limits their utility for training general-purpose robot learning models.

To overcome this limitation, we have re-annotated several real-world robot datasets with more accurate and dense depth information. Specifically, for the DROID dataset (Khazatsky et al., 2025), which provides stereo imagery, we employ FoundationStereo (Wen et al., 2025a) to generate high-quality stereo depth estimates, significantly enhancing the accuracy and density of the depth annotations. For AGIBOT (AgiBot-World-Contributors et al., 2025), the raw depth maps from its native camera are often sparse and of low quality. To address this limitation, we employ PromptDepthAnything (Lin et al., 2025) to perform depth completion, generating dense and high-quality depth information for downstream tasks.

To account for real-world scenarios where high-precision point cloud data may be unavailable, we further utilized MoGe (Wang et al., 2024; 2025a) to re-annotate point clouds for a subset of the BridgeV2 (Walke et al., 2023) and pre-mentioned simulation data. This step allows us to simulate conditions lacking reliable depth information and test our model's robustness.

To improve point cloud downsampling efficiency, we adopt the cropping strategy from 3D Diffusion Policy (Ze et al., 2024). For simulated data, we directly use ground-truth segmentation IDs. For real-world data, we generate foreground masks through a custom pipeline: initial masks from RoboEngine (Yuan et al., 2025) are refined using temporal consistency tracking and integrated with Segment Anything Model 2 (SAM 2) (Ravi et al., 2024) to ensure accuracy and temporal consistency. During both training and inference, we use exclusively uncolored point clouds to ensure the action expert to focus on geometric structures rather than visual semantics. All point clouds are uniformly downsampled to 512 points after crop to maintain a practical balance between spatial accuracy and computational efficiency. Table 12 presents the statistical summary of our dataset. Our dataset will be open-sourced to facilitate research within the community.

*Table 12.* Statistics of Our Pointcloud Datasets

| Environment | Dataset | Number of Trajectories | Arm Type |
|---|---|---|---|
| Simulation | LIBERO (Liu et al., 2023) | 12,000 | Single |
| | RoboTwin (Mu et al., 2025) | 8,000 | Dual |
| | RLBench (James et al., 2019) | 10,000 | Single |
| | CALVIN (Mees et al., 2022) | 20,000 | Single |
| Real World | DROID (Khazatsky et al., 2025) | 76,000 | Single |
| | AGIBOT (AgiBot-World-Contributors et al., 2025) | 10,000 | Dual |
| | BridgeV2 (Walke et al., 2023) | 14,000 | Single |

## B. Details on RoboTwin

We evaluated all the official tasks in RoboTwin, and additionally customized several tasks by modifying the cameras and assets in the simulation environment to assess the generalization ability of our model. Official task definition is listed in B.1, the specific definitions are described in B.2:

### B.1. RoboTwin Official Tasks

**Click Bell**  Press the bell actuator at the indicated position to produce a ringing signal.

**Grab Roller**  Reach for and grasp the roller object securely, then lift or move it to the target location.

**Handover Mic**  Grasp the microphone and transfer it to another agent or specified handover position.

**Lift Pot**  Grasp the pot handle(s) and lift it carefully to the required height or location.

**Place A2B Left**  Pick up object A and place it at location B on the left-side target as specified.

**Place Bread Basket**  Place the bread item into the basket, arranging it neatly.

**Place Phone Stand**  Place the phone onto the stand in the correct orientation.

**Stack Block Two**  Stack two blocks as required and confirm alignment.

**Stack Block Three**  Stack three blocks vertically in the specified order and ensure stability.

**Blocks Ranking RGB**  Rank blocks by visual RGB features and sort them according to the specified criterion.

**Blocks Ranking Size**  Rank blocks by size and arrange them in the required order.

## B.2. RoboTwin Custom Tasks

**Stack Blocks Two Inverse**  Using a previously known instruction, stack two blocks in the reversed sequence. Both the execution order and the instruction order are inverted.

**Stack Random Color Blocks**  Given a novel instruction specifying a color order, stack two blocks of the indicated colors in the required arrangement.

**Move Block Random Color Pad**  Pick up the multicolored block and place it onto the pad with the matching color, ensuring correct pairing and a stable final pose.

**Stack Numberblocks Two**  Select the two designated number blocks and stack them according to their numerical order, from lower to higher.

**Stack Random Blocks Two From Three**  From a set of three available blocks, randomly choose any two and stack them according to the specified color order.

**Place Block Flagpad**  Pick the target block and accurately place it onto the pad marked with a flag, ensuring it is centered and fully supported.

**Move Colorfulblock Colorfulpad**  Pick up the colorful block and place it onto the pad with the corresponding color pattern, aligning edges for a stable placement.

**Place Block Flagpad (choice)**  Choose one of multiple flag-labeled pads and place the block on the selected pad.

**Place A2B Randomly**  Pick an object from region A and place it at a valid random pose within region B, remaining within bounds and avoiding collisions.

## B.3. Generalization Results on RoboTwin

A critical limitation of conventional Vision-Language-Action (VLA) models lies in their reliance on limited training data diversity, which often leads them to memorize specific trajectories rather than learning generalizable skills. This issue becomes particularly evident when these models encounter novel camera viewpoints, unfamiliar scenes, or unseen objects, resulting in significant performance degradation. Our model effectively mitigates this problem. To systematically evaluate its generalization capability, we conducted comprehensive tests across three distinct settings:

1. **Novel Camera Viewpoints:** Using the model trained for our main results, we evaluated its performance on previously unseen camera angles that were absent from the training set. As shown in Table 13, comparison between original and novel viewpoints demonstrates minimal to negligible performance drop.

2. **Novel Objects and Instructions:** We tested the same model on unseen colors, object types, and natural language instructions not present in the training dataset. The results, presented in Table 14, confirm our model's strong generalization capability to novel visual and semantic concepts.

3. **Cross-Environment Transfer:** We further validated our model on the ManiSkill simulation environment, noting that neither the VLM nor the GAE components were trained on any ManiSkill data. Remarkably, as shown in Table 15, our model achieves performance surpassing even the expert-level baseline, demonstrating exceptional cross-environment transferability.

*Table 13.* Camera View Generalization. Our method generalizes to novel cameras with minimal performance drop compared to in-domain settings.

| Task Name | Camera View | |
| --- | --- | --- |
| | **In-Domain** | **Out-of-Domain** |
| Place A2B Left | 0.52 | 0.51 |
| Place A2B Right | 0.48 | 0.46 |
| Stack Block 3 | 0.36 | 0.30 |
| Stack Bowl 3 | 0.54 | 0.56 |
| Rank Size | 0.52 | 0.49 |
| Rank RGB | 0.71 | 0.68 |

*Table 14.* Zero-shot Results. Performance comparison on novel colors, objects, and language instructions.

| Category | Task | Ours | Pi0 |
| --- | --- | --- | --- |
| **Unknown Color** | Stack Block | **0.86** | 0.12 |
| | Rank RGB | **0.69** | 0.32 |
| **Novel Object** | Stack Number | **0.38** | 0.00 |
| | Click Clock | **0.58** | 0.20 |
| **Semantic** | Place Right | **0.39** | 0.23 |
| | Place Random | **0.28** | 0.16 |

*Table 15.* ManiSkill Zero-Shot Performance. Our model demonstrates strong direct transfer capabilities to new simulation environments.

| Task | Ours | DP3 | ACT |
| --- | --- | --- | --- |
| Push Cube | **0.89** | 0.83 | 0.81 |
| Stack Cube | **0.84** | 0.76 | 0.69 |
| Pull Cube | **0.62** | 0.48 | 0.40 |

# C. Real World Robot Setting

Our experimental platform consists of both single-arm and dual-arm real-world robotic systems. The single-arm setup is centered around a 7-DoF Franka Research 3 robotic arm equipped with a UMI gripper (Chi et al., 2024) for versatile object manipulation. The dual-arm setup uses an AgileX PiperX bimanual robot, where each arm is independently actuated and equipped with a parallel-jaw gripper, enabling coordinated bimanual manipulation.

For visual perception, a RealSense D435 RGB-D camera is statically mounted in a third-person viewpoint overlooking the workspace, capturing images at a resolution of $640 \times 480$. For the single-arm system, expert demonstrations are collected via teleoperation using a 6-DoF 3D mouse, adapted from a publicly available implementation[1] . For the dual-arm system, demonstrations are collected using a Pico-based teleoperation interface, which allows an operator to simultaneously control both arms in Cartesian space.

---

[1]https://github.com/UT-Austin-RPL/deoxys_control

The robot control loop runs at 20 Hz, obtained by down-sampling from the native 100 Hz controller to balance temporal smoothness and data efficiency. The action space is defined in SE(3), where each action is a 7-dimensional vector specifying an absolute target end-effector pose, consisting of a 3D Cartesian position and a 4D quaternion orientation.

To rigorously evaluate our method, we design a benchmark suite comprising ten manipulation tasks spanning both single-arm and dual-arm settings. The single-arm benchmark includes six tasks, covering short-, middle-, and long-horizon manipulation. The dual-arm benchmark contains four tasks that emphasize bimanual coordination and contact-rich interactions. Across both settings, tasks are further categorized into short-horizon, middle-horizon, and long-horizon groups, enabling systematic evaluation of fine-grained visuomotor control, multi-stage reasoning, and long-horizon sequential planning.

For each task, we adopt a unified data collection and evaluation protocol. We collect 20 expert demonstration trajectories per task for training. During evaluation, each trained policy is tested for 20 independent trials with randomized initial conditions, providing a robust estimate of task success and generalization. The detailed task definitions and objectives are described below.

**1) pick up toy:** The robot must grasp a specific target toy from the tabletop. This task tests basic object identification and manipulation.

**2) open drawer:** The robot is required to interact with an articulated object by approaching a closed drawer and pulling its handle to open it.

**3) stack cubes:** The robot needs to precisely pick up a cube and place it on top of one another.

**4) put toy into basket:** This is a pick-and-place task where the robot must first pick up a specified toy and then deposit it into a nearby basket.

**5) rank cubes:** The robot must perceive a specific attribute of several cubes, such as size or color, and arrange them in a designated sequence.

**6) stack cups:** The robot's objective is to stack several cups of varying sizes in descending order, requiring it to place the largest cup first and nest the smaller ones inside it.

**7) stack the yellow bowl on the green bowl:** The robot's objective is to grasp the yellow bowl and place it stably on top of the green bowl, forming a correct stacked configuration.

**8) erase the triangle symbol on the whiteboard:** The robot's objective is to use the eraser to remove a triangle symbol from the whiteboard, requiring accurate positioning and surface contact.

**9) fold the towel twice with dual arms:** The robot's objective is to collaboratively manipulate a towel with both arms and fold it twice into a compact configuration.

**10) insert the flower into the vase:** The robot's objective is to grasp the flower and insert its stem into the opening of the vase without collision.

**Out-of-Distribution (OOD) Setup.**

As shown in Figure 7, to systematically evaluate position generalization, we discretize the tabletop workspace into a $7 \times 10$ grid using a localization light projector, where each illuminated grid cell corresponds to a unique object placement position. A subset of grid locations is used to define the In-Distribution (ID) positions during training, while OOD positions are selected from the remaining grid cells that never appear in the training set. For each task, we choose five unseen OOD positions and evaluate the policy four times at each position. This grid-based protocol enables precise control of object placement and provides a clear separation between ID and OOD spatial configurations.

## D. Training Details

The Vision-Language Model (VLM) is fine-tuned for 1,000 steps with a batch size of 32 on eight 80GB NVIDIA A100 GPUs. The Generalizable Action Expert (GAE) is trained on the same hardware configuration using our two-stage training paradigm. Specifically, the action pre-training stage runs for 8 hours with a batch size of 32,768 using trajectory-only data, enabling efficient large-scale learning of basic motion skills. This is followed by a point cloud fine-tuning stage that runs for 2 days with a batch size of 256, where high-quality point cloud observations are introduced to learn geometry-aware refinement. Adam optimizer is used for both stages with the learning rate and other hyperparameters detailed in Table 16.

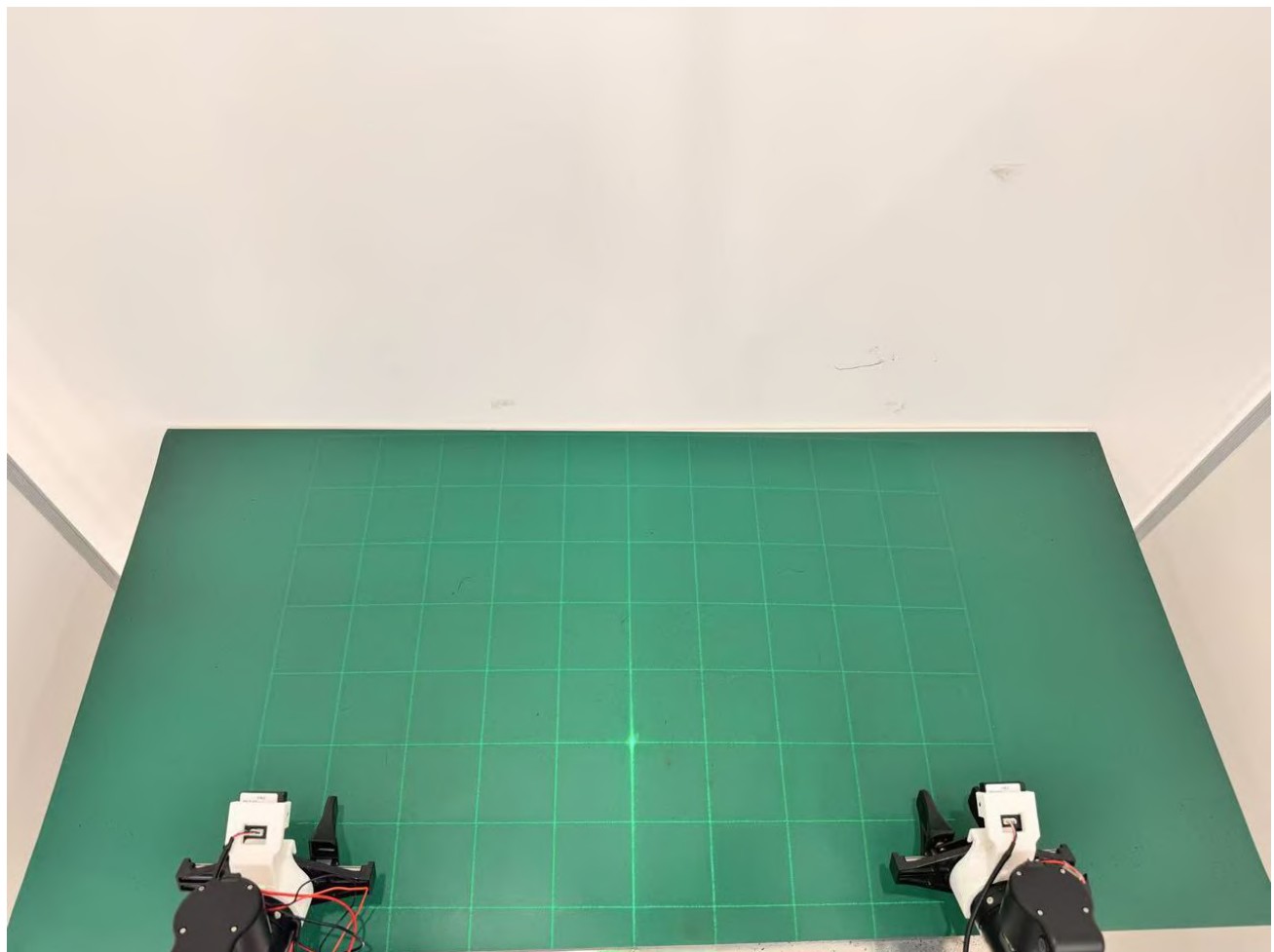

*Figure 7.* We discretize the tabletop workspace into a $7 \times 10$ grid using a localization light projector, where each illuminated grid cell corresponds to a unique object placement position

## E. Visualization Results

This subsection presents our simulation and physical experiment results.

### E.1. Simulation Robot Results

Figure 9 shows our demonstrations in the RoboTwin (Chen et al., 2025) simulation environment. Our model exhibits strong robustness under varying lighting conditions, different scene configurations, and previously unseen objects.

### E.2. Real World Robot Results

Figure 10 and Figure 11 shows demonstrations of our model operating in the real world. Our model exhibits strong performance across tasks with different horizons, and is particularly effective on long-horizon tasks.

## F. More Ablation Study

### F.1. Comparision of Trajectory Planning Quality

The comparison in Table 18 evaluates the performance of our VLM model's planning trajectory in 2D against three representative baselines: Magma (Yang et al., 2025), HAMSTER (Li et al., 2024b), and LLARVA (Niu et al., 2024). The evaluation covers 11 metrics that collectively measure coverage accuracy, geometric similarity, and fine-grained spatial

*Table 16.* **Training Hyperparameters for the Generalizable Action Expert (GAE).**

| Hyperparameter | Value |
|---|---|
| Backbone | Conv-based Diffusion Policy |
| Optimizer | AdamW |
| Learning Rate | 1e-4 |
| Weight Decay | 1e-6 |
| Batch Size (Pre-training) | 32,768 |
| Batch Size (Pointcloud Finetuning) | 256 |
| Training Time | 8 hours (Pretrain) + 2 days (Finetune) |
| Input Modality | Trajectory / Point Cloud |
| Point Cloud Size | 512 points |
| Action Representation | End-effector Pose |
| Noise Steps (Train / Inference) | 500 / 10 |
| Task-specific Finetuning | None |
| Hardware | $8 \times$ NVIDIA A100 (80GB) |

*Table 17.* **SimplerEnv evaluation across different models on WidowX Robot tasks.**

| Model | Put Spoon on Towel | | Stack Green on Yellow | | Put Carrot on Plate | | Put Eggplant in Basket | | Overall Average | |
|---|---|---|---|---|---|---|---|---|---|---|
| | Grasp Spoon | Success | Grasp G Block | Success | Grasp Carrot | Success | Grasp Eggplant | Success | Grasp Avg. | Success Avg. |
| RT-1-X (Brohan et al., 2022) | 16.7% | 0.0% | 8.3% | 0.0% | 20.8% | 4.2% | 0.0% | 0.0% | 11.5% | 1.1% |
| Octo-Base (Team et al., 2024) | 34.7% | 12.5% | 31.9% | 0.0% | 52.8% | 8.3% | 66.7% | 43.1% | 46.5% | 16.0% |
| SpatialVLA (Wu et al., 2025a) | 25.0% | 20.8% | 58.3% | 25.0% | 41.7% | 20.8% | 79.2% | 70.8% | 51.1% | 34.4% |
| $\pi_0$ (Black et al.) | 45.8% | 29.1% | 25.0% | 0.0% | 50.0% | 16.7% | 91.6% | 62.5% | 40.1% | 27.1% |
| $\pi_0$-FAST (Pertsch et al., 2025) | 62.5% | 29.1% | 58.5% | 21.9% | 54.0% | 10.8% | 83.3% | 66.6% | 48.3% | 32.1% |
| OpenVLA (Kim et al., 2024) | 4.1% | 0.0% | 33.0% | 0.0% | 12.5% | 0.0% | 8.3% | 4.1% | 7.8% | 1.1% |
| GR00T-N1 (Bjorck et al., 2025) | 83.3% | 62.5% | 54.2% | 45.8% | 70.8% | 16.7% | 41.7% | 20.8% | 49.5% | 36.5% |
| UniVLA (Bu et al., 2025) | 76.4% | 52.8% | 66.7% | 2.8% | **79.2%** | 55.6% | 87.5% | 66.7% | 77.5% | 45.6% |
| SoFAR (Qi et al., 2025) | 62.5% | 58.3% | 91.7% | 70.8% | 75.0% | **66.7%** | 66.7% | 37.5% | 74.0% | 58.3% |
| **Ours** | **95.8%** | **79.2%** | **93.0%** | **72.1%** | 76.8% | 57.4% | **100.0%** | **91.7%** | **91.4%** | **75.1%** |

alignment. As indicated by the notation in the table, higher values correspond to better performance for cover F1, cover precision, cover recall, and LCSS similarity. Lower values correspond to better performance for all distance-based metrics such as DTW, endpoint error, and Frechet distance, since these metrics quantify deviations from the target trajectory and smaller values indicate closer alignment with the ground truth.

Across all metrics, our VLM achieves the strongest performance among the four methods. For the coverage metrics, our method obtains the highest values, including a cover F1 of 0.9546, which surpasses Magma at 0.9282, HAMSTER at 0.9451, and LLARVA at 0.9189. Our cover precision is likewise the highest at 0.9558, and our cover recall reaches 0.9550, indicating that our method captures more of the target shape and does so with fewer errors compared to the baselines.

For distance-based and discrepancy metrics, our VLM again performs best. The DTW score of 0.4564 is substantially lower than the corresponding values for Magma (0.8353), HAMSTER (0.7945), and LLARVA (0.4894). Endpoint error shows a similar improvement, with our value of 0.0739 clearly below Magma (0.1784), HAMSTER (0.1181), and LLARVA (0.0771). Frechet distance follows the same trend, and our score of 0.1035 remains the lowest among all methods.

For fine-grained spatial metrics such as max, mean, and median orthogonal distances, our results consistently show the smallest deviations from the target trajectory. Although LLARVA performs relatively well in some endpoint-related metrics, its orthogonal distances, for example a mean orthogonal distance of 0.0677, remain noticeably higher than the value obtained by our VLM, which is 0.0322. The same pattern appears for startpoint error and LCSS similarity, where our method demonstrates the closest overall alignment and the most consistent trajectory reconstruction.

**Inference Speed.** While conventional VLAs require a full backbone pass for each inference step, our architecture allows the VLM to reason an entire trajectory segment at once. This is then refined in real-time by the GAE into executable trajectories at a significantly higher frequency. Numerically, while OpenVLA runs at 4.63 Hz on a single RTX 4090, our system achieves a rate of 11.43 Hz.

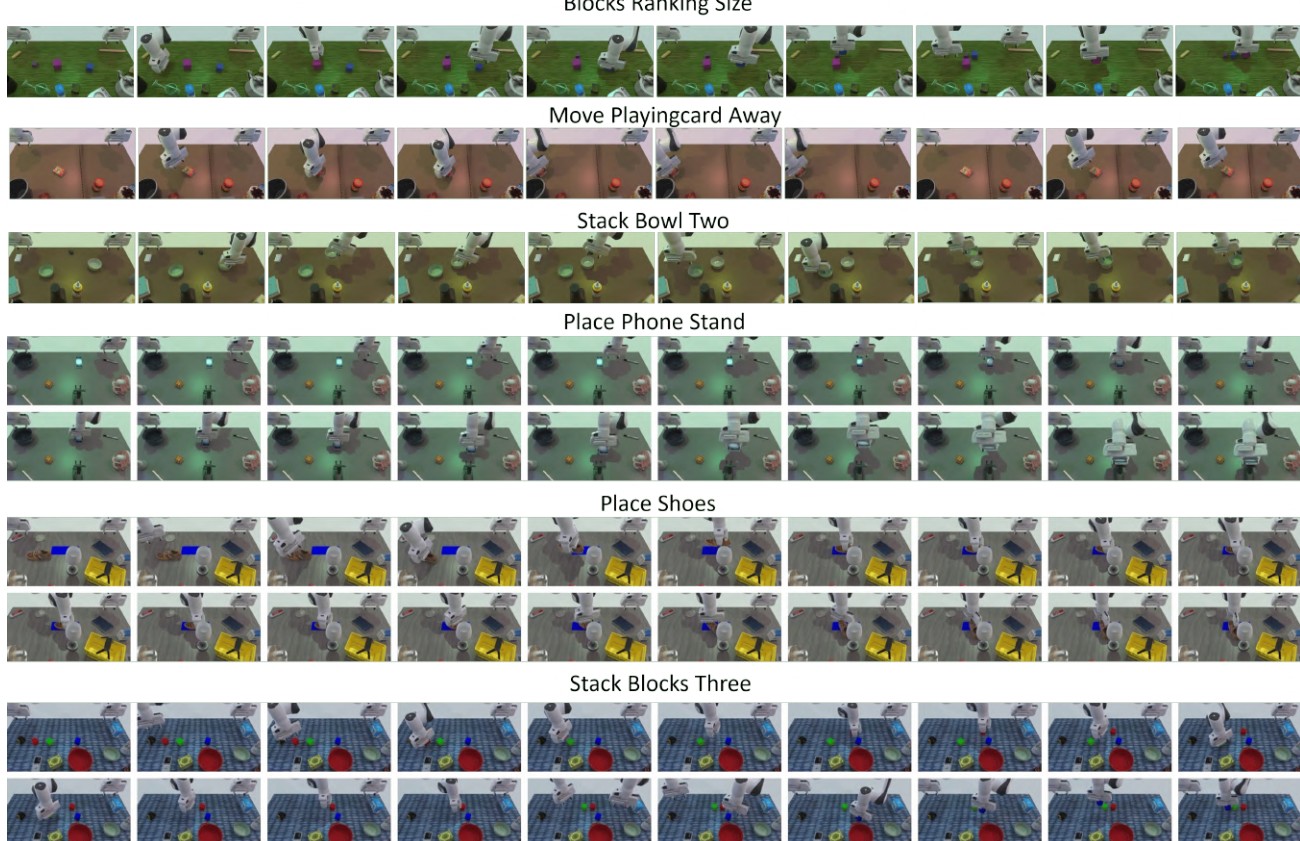

*Figure 8.* Simulation Robot Results: PnP tasks

## F.2. Ablation on Different Depth Noise Levels

We evaluate the model's robustness to depth noise by running each task 20 times under different noise levels and reporting the percentage success rate, results are shown in Table 19. The results reveal three clear trends. **(1) Robustness under low noise**: When the injected depth noise is small (approximately 1 cm), performance remains close to the noise-free baseline. **(2) Degradation under moderate noise**: At around 10 cm noise, most tasks exhibit a noticeable decline in success rates. **(3) Failure under high noise**: With extreme noise levels (around 1 m), the system almost completely fails.

Overall, although depth noise inevitably affects performance, the model remains stable within the typical error range of commodity depth sensors (1–10 cm). Interestingly, for tasks resembling the training distribution, the model can still function under significant noise, implying that it has implicitly learned coarse depth-estimation priors. However, complex scenes still require more accurate depth inputs due to the ill-posed nature of depth estimation.

Compared with 3D-dense approaches that depend on full point clouds, our method only requires one or two sparse depth anchors to calibrate the VLM's spatial reasoning. Modern single-point ranging modules (e.g., SK60, ±2 mm accuracy) make this requirement lightweight and practical for hardware deployment.

## F.3. Ablation on Different Density of Waypoints

To investigate the effect of waypoint density on model performance, an ablation study was conducted using different training settings: 5 points (ours), 15 points, and 30 points, with results summarized in the Table 20. This analysis highlights a critical trade-off where increasing control fidelity via denser waypoints significantly undermines the model's pre-trained general capabilities. A clear trend emerges when examining the Logic performance metrics: as the waypoint density increases, the model's general knowledge exhibits continuous deterioration, a classic symptom of catastrophic forgetting. The baseline Qwen 2.5VL 7B model starts with high scores (C-eval: 76.15), but scores progressively drop to 63.45 (5 points), 61.16 (15

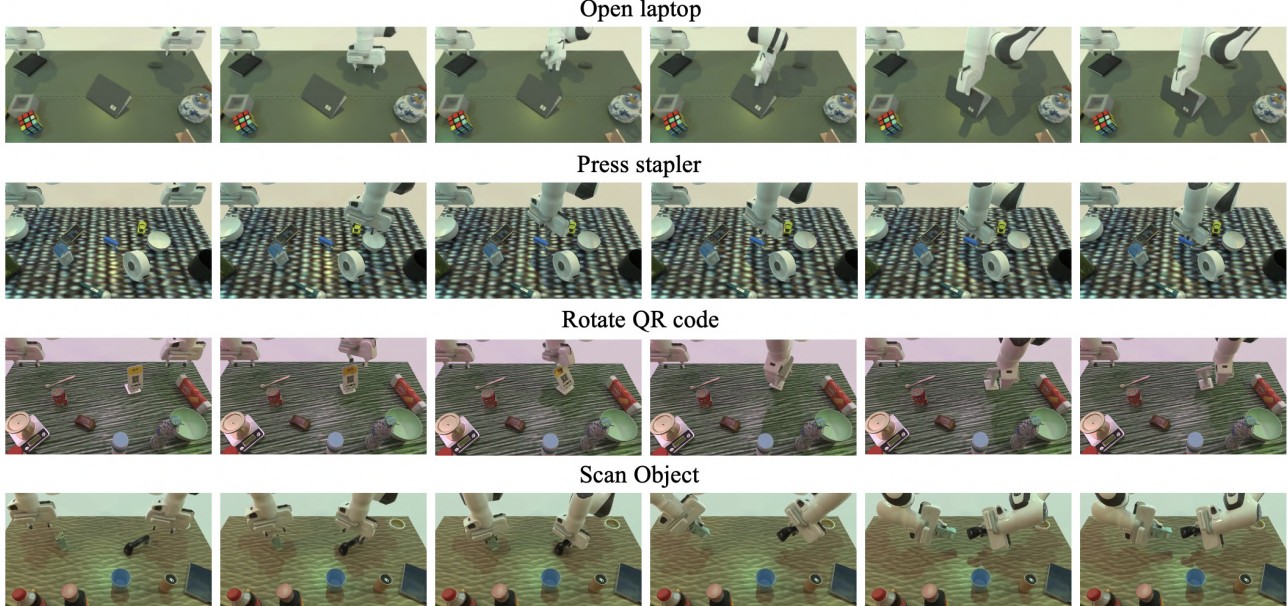

*Figure 9.* Simulation Robot Results: non-PnP tasks

points), and 58.45 (30 points). This strong, inverse correlation suggests that introducing a higher frequency of action control signals conflicts severely with the model's foundational language representations.

Crucially, while the model's general ability declines, the Task performance (success rate) does not exhibit a corresponding upward trend. Comparing the sparse 5 points setting with the denser 30 points setting, performance is often comparable or even slightly worse at the higher density. For instance, the success rate for place_a2b_right drops from 0.30 to 0.25, and blocks_ranking_size drops from 0.55 to 0.35. This imbalance suggests that the current approach adopted by Vision-Language-Action (VLA) models, which often relies on dense trajectory prediction, contains substantial room for compression and optimization.

These findings imply that much of the information within the dense action sequences is either redundant for achieving key task goals or introduces detrimental noise that degrades the model's ability to generalize and reason. Therefore, focusing on sparse, kinematics-based keyframes, such as those used in the 5 points setting, provides sufficient, high-value control information to successfully execute tasks while effectively protecting the foundational pre-trained knowledge.

### F.4. Ablation on Different Models and Different Model Sizes

In the paper, we utilized Qwen 2.5VL 7B (Bai et al., 2025) as our base Vision-Language Model (VLM). However, adopting other models is also very convenient, as we adhere to a unified output format. This format allows other models to be easily fine-tuned under frameworks like LlamaFactory (Zheng et al., 2024), enabling them to acquire the capability for physical world manipulation. We compared different sizes of Qwen models (Bai et al., 2025) as well as the InternVL (Chen et al., 2024b) model, and the results are presented in the Table 21.

*Table 21.* Ablation study of different training strategy.

|  | Short Horizon | Middle Horizon | Long Horizon |
| --- | --- | --- | --- |
| **Qwen2.5VL 7B** | **0.82** | **0.76** | **0.60** |
| **Qwen2.5VL 3B** | 0.76 | 0.58 | 0.39 |
| **InternVL3 8B** | 0.80 | 0.64 | 0.54 |
| **InternVL3 4B** | 0.69 | 0.55 | 0.37 |

*Table 18.* Comparison of performance metrics across different methods.

| Metric | Ours | Magma (Yang et al., 2025) | HAMSTER (Li et al., 2024b) | LLARVA (Niu et al., 2024) |
|---|---|---|---|---|
| cover f1 ($\uparrow$) | **0.9546** | 0.9282 | 0.9451 | 0.9189 |
| cover precision ($\uparrow$) | **0.9558** | 0.9484 | 0.9504 | 0.9201 |
| cover recall ($\uparrow$) | **0.9550** | 0.9534 | 0.9440 | 0.9415 |
| dtw ($\downarrow$) | **0.4564** | 0.8353 | 0.7945 | 0.4894 |
| endpoint err ($\downarrow$) | **0.0739** | 0.1784 | 0.1181 | 0.0771 |
| frechet ($\downarrow$) | **0.1035** | 0.2296 | 0.1311 | 0.1673 |
| hausdorff ($\downarrow$) | **0.0809** | 0.1282 | 0.0822 | 0.0903 |
| max orth dist ($\downarrow$) | **0.0744** | 0.1106 | 0.1232 | 0.0923 |
| mean orth dist ($\downarrow$) | **0.0322** | 0.0476 | 0.0356 | 0.0677 |
| median orth dist ($\downarrow$) | **0.0286** | 0.0439 | 0.0410 | 0.0532 |
| startpoint err ($\downarrow$) | **0.0778** | 0.1449 | 0.1411 | 0.1743 |
| lcss sim ($\uparrow$) | **0.9660** | 0.9562 | 0.9496 | 0.9447 |

*Table 19.* Task performance under different noise levels.

| Task / Noise Level | 0.00 | 1.00 | 10.00 | 100.00 |
|---|---|---|---|---|
| place_a2b_left | 0.40 | 0.35 | 0.20 | 0.10 |
| place_a2b_right | 0.30 | 0.25 | 0.15 | 0.05 |
| place_object_scale | 0.45 | 0.25 | 0.25 | 0.15 |
| stack_blocks_three | 0.40 | 0.30 | 0.00 | 0.00 |
| stack_bowl_three | 0.80 | 0.40 | 0.10 | 0.10 |
| blocks_ranking_size | 0.55 | 0.20 | 0.00 | 0.00 |
| blocks_ranking_rgb | 0.65 | 0.35 | 0.05 | 0.10 |

*Table 20.* Ablation on Different Density of Waypoints

| Metric | Qwen 2.5VL 7B | 5 points (ours) | 15 points | 30 points |
|---|---|---|---|---|
| **Logic performance** | | | | |
| ceval avg. | 76.15 | 63.45 | 61.16 | 58.45 |
| cmmlu avg. | 75.11 | 63.88 | 61.41 | 59.36 |
| mmlu avg. | 70.08 | 61.69 | 60.63 | 60.11 |
| **Task performance** | | | | |
| place_a2b_left | - | 0.40 | 0.35 | 0.40 |
| place_a2b_right | - | 0.30 | 0.20 | 0.25 |
| place_object_scale | - | 0.45 | 0.50 | 0.40 |
| stack_blocks_three | - | 0.40 | 0.45 | 0.35 |
| stack_bowl_three | - | 0.80 | 0.70 | 0.80 |
| blocks_ranking_size | - | 0.55 | 0.30 | 0.35 |
| blocks_ranking_rgb | - | 0.65 | 0.55 | 0.55 |

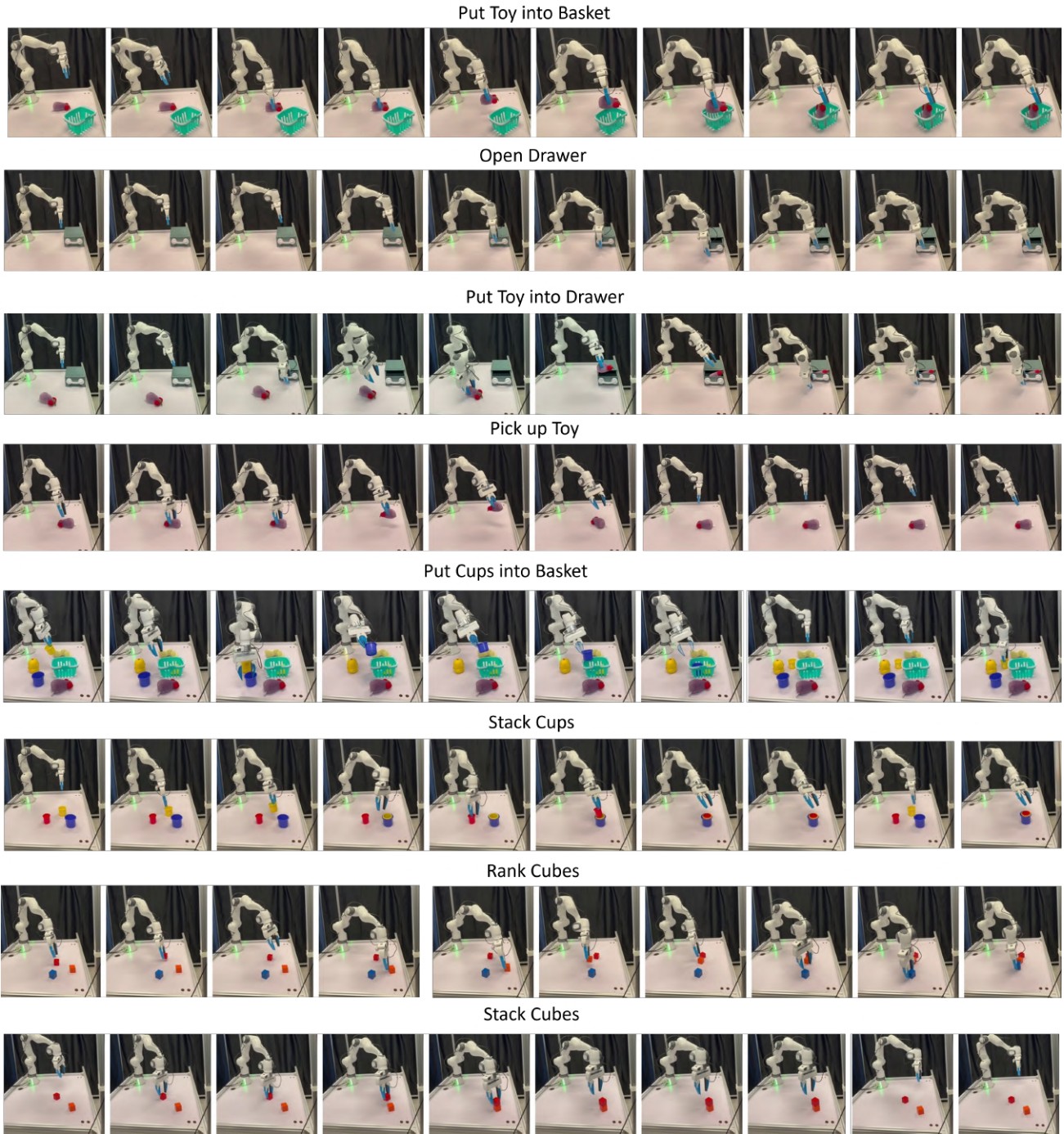

*Figure 10.* Real World Single Arm Robot Results.

stack the yellow bowl on the green bowl

erase the triangle symbol on the whiteboard

fold the towel twice with dual arms

insert the flower into the vase

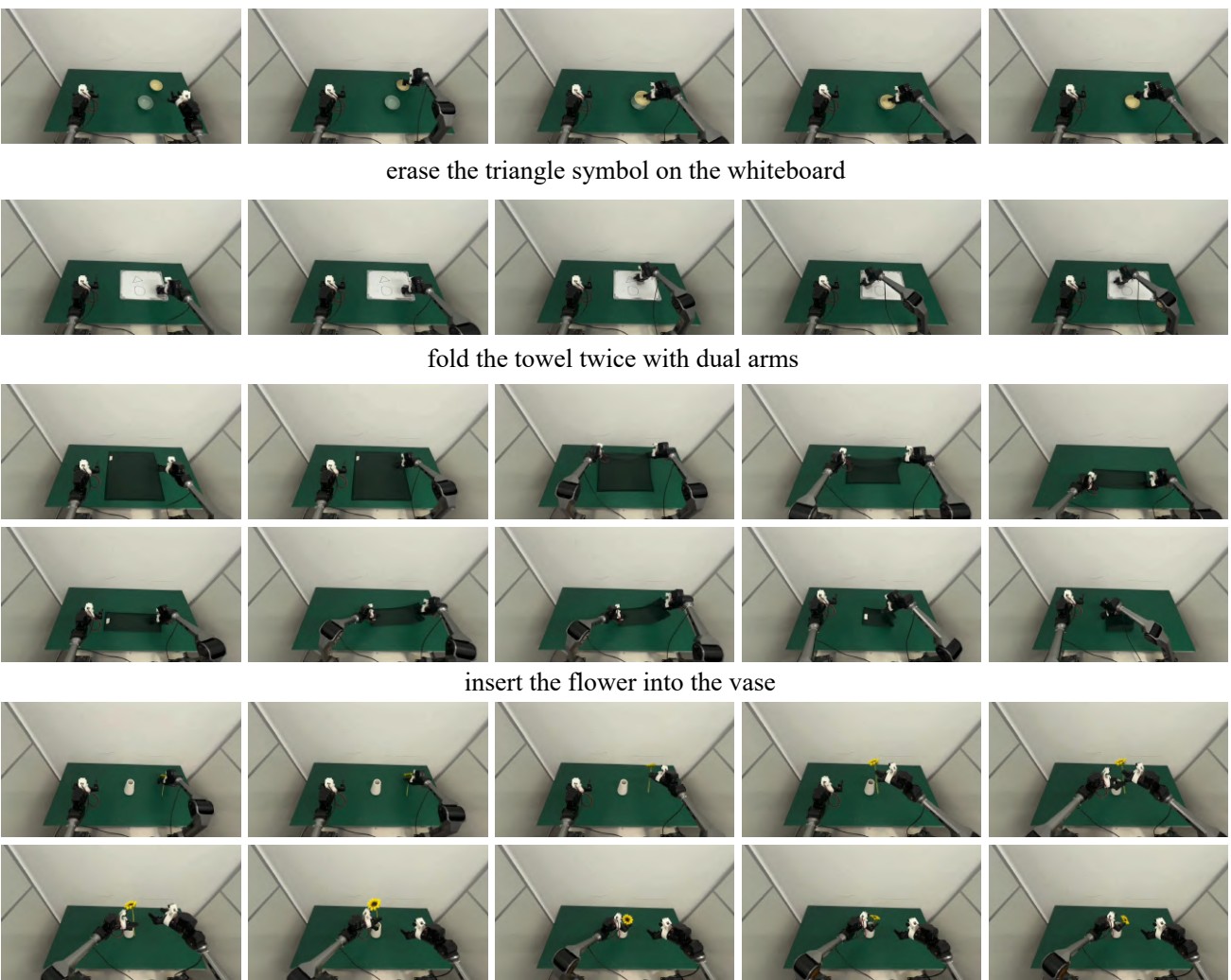

*Figure 11.* Real World Dual Arm Robot Results.

