# OpenReview forum: "GAE: Unleashing Physical Potential of VLM with Generalizable Action Expert"
_ICML.cc/2026/Conference — ICML 2026 regular_

### Official Review · Reviewer_4rNU · 2026-03-09

**Soundness:** 4
**Presentation:** 3
**Significance:** 3
**Originality:** 3
**Overall Recommendation:** 6
**Confidence:** 3

**Summary:**

The paper presents a practical hierarchical VLA framework that combines high-level semantic planning with low-level geometric action generation, and its main strength is that it offers a modular and effective design for improving robot control across both simulated and real-world settings.

**Compliance With Llm Reviewing Policy:**

Affirmed.

**Final Justification:**

addressed my main concerns

**Key Questions For Authors:**

Can the authors clarify what exact evidence supports the claim that the proposed decomposition reduces catastrophic forgetting, beyond reducing the amount of fine-tuning needed for downstream performance?

**Limitations:**

The paper does not yet convincingly show that the sparse 3D pose representation is a principled choice, rather than one reasonable design option among several possible intermediate interfaces.

**Strengths And Weaknesses:**

Soundness:
The method is reasonable, but the evidence is not yet strong enough to support several of the main claims. In particular, the paper suggests that the proposed decomposition helps reduce catastrophic forgetting and better preserve the VLM’s general capabilities, but the current results do not really prove that. The MMLU numbers mainly show that the model may need fewer fine-tuning steps, which is not the same as showing less forgetting under a controlled comparison. Also, some of the baselines are not fully comparable because they differ in pretraining data, training setup, and fine-tuning regime. On top of that, the experiments would be more convincing with stronger statistical reporting, such as multiple seeds, confidence intervals, or significance tests.

Presentation:
The paper is readable overall, and the main idea is not hard to follow, but the writing still needs to be sharper. The key story around using a sparse 3D interface to separate high-level reasoning from low-level control is intuitive, but the paper does not explain clearly enough why this specific interface is the right choice or how it differs in a principled way from closely related hierarchical VLA work.

Significance:
This is a useful direction for robot learning, and the modular setup could be practically helpful in settings where end-to-end action supervision is expensive or brittle. That said, the current version does not yet make a strong enough case for broad significance. Right now, the work feels more like a solid systems effort than a result that clearly changes how people will think about learning-based robot control. It could still be useful to practitioners, but the paper would need a stronger learning insight or a more clearly validated abstraction to make a stronger case here.

Originality:
There are some interesting pieces here, especially the sparse 3D pose interface and the staged APPF training strategy. However, the overall novelty is still not fully clear. The main design idea, a high-level planner plus a lower-level geometric controller, is already quite close to several recent hierarchical VLA works.

---

> ### Author Rebuttal · Authors · 2026-03-30
>
> We thank the reviewer for the careful and constructive feedback. We respond to the main concerns below.
> >💬 **W1 & Q1 & Key Questions:** *Evidence for reduced catastrophic forgetting.*
>
> We thank the reviewer for this important question. We agree that this claim should be interpreted carefully.
>
> **First**, we believe the fact that our method reaches strong downstream performance with substantially fewer finetuning steps is a suggestive signal. Catastrophic forgetting is often associated with task-specific parameter updates that may overwrite pretrained representations. From this perspective, requiring fewer updates is at least consistent with the possibility that the pretrained VLM is perturbed less. This interpretation is also broadly consistent with our MMLU results, where heavier finetuning leads to degraded performance on a benchmark of general language and reasoning ability.
>
> **Second**, this interpretation is supported by the design analysis in our paper. Sparse waypoint supervision acts as a compressed target that preserves task-relevant information while discarding redundant embodiment-specific control details; it also concentrates gradient on a small number of decision-critical points instead of spreading it across many temporally correlated low-information action steps. Moreover, our VLM predicts **image-/camera-aligned anchors** and represents actions directly in **text**, keeping the prediction space much closer to the pretrained visual-semantic space than dense action prediction.
>
> > 💬 **W2 & Q2 & Limitations:** *Presentation.*
>
> We thank the reviewer for this important point. We agree that the paper should explain more clearly why the sparse 3D pose interface is a principled design choice. Our motivation is not that sparse 3D pose is the only possible intermediate representation, but that it is particularly well matched to the role of a reusable low-level action expert. Specifically, the interface is designed to be: **(i) geometric**, so that it can directly guide motion refinement; **(ii) sparse**, so that the high-level VLM only needs to provide task-critical guidance rather than dense low-level control; and **(iii) minimal**, so that most semantic burden is removed from the low-level controller. This allows GAE to focus on geometry-aware motion generation, rather than parsing semantically rich intermediate representations, and is exactly what makes cross-task reuse possible.
>
> Compared with many hierarchical VLA designs, the key distinction is therefore not decomposition alone, but the choice of an interface that is intentionally lightweight and action-relevant for the low-level expert. We will make this design rationale more explicit in the revision.
>
> **On the limitation regarding whether sparse 3D pose is the optimal interface.**
>
>  We agree that the current paper does not establish sparse 3D pose as the uniquely optimal interface, and we do not intend to make that claim. More precisely, we view it as a validated and effective design choice that is sufficiently minimal, geometric, and executable to support a reusable action expert. We see this as a framework-level contribution: the main goal of this work is to demonstrate that the action expert itself can be scaled into a reusable module, rather than to exhaustively identify the best possible intermediate representation. We believe this provides a strong baseline for future work to explore alternative interfaces.
>
> Possible directions include richer but still compact geometric representations, such as structured keypoint sets, trajectory tokens with explicit spatial grounding, implicit latent motion representations, or learned continuous interfaces that preserve executability while further reducing information loss.
>
> > 💬 **W3 & Q3:** *Originality: Difference from recent hierarchical VLA works.*
>
> We agree that the high-level planner plus low-level controller paradigm itself is not new, and this is not the main novelty we intend to claim. Our contribution is instead the explicit validation that the **action expert itself can be made general, reusable, and scalable**. In many recent hierarchical VLA systems like HAMSTER, BridgeVLA, ThinkAct, and GR00T N1, the emphasis remains on scaling the high-level VLM, while the low-level controller is still task-dependent, tightly coupled to upstream representations, or adapted for new tasks. As a result, these works do not fully demonstrate that the low-level expert can be trained once and broadly reused across tasks.
>
> In contrast, our work is built around a **generalizable action expert (GAE)** that transfers across tasks without task-specific finetuning. The sparse 3D pose interface is not introduced as an arbitrary intermediate representation, but as a means to remove most semantic burden from the low-level controller, so that GAE can focus on geometry-aware motion refinement and become reusable across tasks.

---

> > ### Author Rebuttal · Reviewer_4rNU · 2026-03-31
> >
> > Thank you for the thoughtful rebuttal. Your response clarifies the intended scope of the catastrophic forgetting claim and explains more clearly why the sparse 3D pose interface is a deliberate, lightweight, and reusable design choice rather than an arbitrary intermediate representation. I am satisfied that my main concerns have been addressed, especially regarding the framing of the contribution and the role of the generalizable action expert.

---

### Official Review · Reviewer_motz · 2026-03-10

**Soundness:** 3
**Presentation:** 3
**Significance:** 3
**Originality:** 3
**Overall Recommendation:** 4
**Confidence:** 4

**Summary:**

This paper proposes a task-agnostic model that can serve as a Generalizable Action Expert (GAE) to tackle the generalized manipulation problem with a finetuned VLM. In contrast to most current methods that directly build an end-to-end optimized VLA, this paper builds a dataset and explores to learn a diffusion policy conditioned on 3D sparse pose trajectories and point clouds of the environment thereon using an Action Pre-training, Pointcloud Fine-tuning (APPF) strategy. Simulation and real-world experiments show the promising results of this approach. Extensive ablation studies show the efficacy of detailed designs.

**Compliance With Llm Reviewing Policy:**

Affirmed.

**Final Justification:**

I would keep my initial score. See the detailed justification in the reply during the discussion phase.

**Key Questions For Authors:**

See weaknesses.

Another minor question is what the setup is for VLM fine-tuning for different experiments. How about the results of finetuning VLM in a multi-task, multi-benchmark manner vs. specialized finetuning? How to make a fair comparison with the other methods under such a setup?

**Limitations:**

Yes.

**Strengths And Weaknesses:**

Strengths:

- The idea is novel. This paper explores a pathway to build generalizable manipulation polices that further decouple the control part from semantic understanding, reasoning, and motion planning.

- The paper contributes both a dataset for research along this line as well as comprehensive validation.

- The overall writing is smooth with clear motivation and presentation.

- The simulation and real-world results seem to be promising. Extensive ablation studies analyze the benefits of implementation details.

- The action experts are generalizable and perform in a zero-shot manner in all the experiments.


Weaknesses:

- The real-world demo seems to be very easy, even for the "long-horizon" tasks, making the real generalization capability of this method suspicious.

- The simulation experiments do not compare with the latest state-of-the-art methods, which can surpass the results shown in the paper. Although it is not necessary to compare with arXiv papers, I would recommend that the author make a more solid comparison and discussion with those methods, where there can be subtle tricks that may affect the results a lot.

- There are still several details that can be polished, such as Fig. 1 is too small as a main figure, a lack of implementation details for Sec. 3.1, and the redundant "." in the title of Sec. 3.2 and the missing one in line 267, inappropriate organization of results in Table 6 (mix of methods and metrics).

---

> ### Author Rebuttal · Authors · 2026-03-31
>
> We thank the reviewer for the positive assessment and the constructive suggestions. We address the main concerns below.
> > 💬 **W1 & Q1:** *On the real-world long-horizon evaluation.*
>
> We thank the reviewer for this helpful comment. We agree that the current real-world section could better illustrate the practical generalization of our method. To address this, we provide four additional real-world examples in the anonymous supplementary link: https://anonymous.4open.science/r/73C2/, including **making a hamburger, arranging and pushing a 3-block domino, opening a bag zipper and placing a cloth inside, and stacking/folding two socks**. These examples involve longer execution chains, more diverse object interactions, and richer scene variation. Importantly, in all four cases, GAE is deployed zero-shot, without task-specific finetuning or any additional real-world action-expert training. While these examples are not intended to be exhaustive, we hope they better illustrate the transferability and robustness of the proposed framework in more challenging real-world settings.
>
> > 💬 **W2 & Q2:** *On comparison with recent SOTA methods.*
>
> We thank the reviewer for this valuable suggestion. We agree that recent methods such as **LingBot-VA [1]** and **Motus [2]** should be discussed more explicitly. These methods mainly improve performance by scaling the overall world-action model (WAM) stack, e.g., through larger video-action models, stronger visual priors, and explicit future-state modeling. LingBot-VA jointly models video prediction and action execution in a shared latent space, while Motus is a unified latent action world model integrating understanding, video generation, and action modeling.
>
> Our paper studies a complementary question: rather than maximizing performance by scaling the full world-model stack, we ask whether the action expert itself can be trained as a reusable and scalable module under a clean high-level/low-level decomposition. In this sense, stronger visual representations, larger VLMs, and more powerful reasoning or world-modeling components are largely **orthogonal** to our main contribution. We will make this positioning clearer in the revision and add a more careful discussion of recent WAM methods and their setting differences.
>
> > 💬 **W3 & Q3:** *On VLM finetuning setup and fairness.*
>
> We thank the reviewer for this important question and apologize that the VLM finetuning setup was not described clearly enough in the current draft.
> In our experiments, we use two separate VLMs for simulation and real-world settings. For simulation, the VLM is finetuned using data from SIMPLEREnv and RoboTwin. For real-world experiments, the VLM is finetuned using our collected real-robot data from single-arm Franka and bimanual PiperX. Our default setup in the paper uses multi-benchmark joint finetuning for the simulation VLM, because we want to test whether the high-level model can provide sparse geometric guidance across diverse environments while keeping the low-level GAE fixed and reusable. To address the reviewer’s concern more directly, we additionally finetuned the simulation VLM in a single-benchmark specialized manner on SIMPLEREnv only and RoboTwin only. The comparison is summarized below.
>
> **SimplerEnv**
>
> | Model | Google Robot | WidowX Grasp | WidowX Success |
> |:---:|:---:|:---:|:---:|
> | Multi-bench | 74.1% | 91.4% | 75.1% |
> | Single-bench | **74.8%** | **91.7%** | **75.8%** |
>
> **RoboTwin**
>
> | Model | Short Horizon | Middle Horizon | Long Horizon |
> |:---:|:---:|:---:|:---:|
> | Multi-bench | 0.82 | 0.77 | **0.60** |
> | Single-bench | **0.85** | **0.78** | 0.58 |
>
> As expected, single-benchmark finetuning performs slightly better on its own test benchmark. This is also the setting followed by the other baselines in our paper, so such gains are reasonable. However, this is exactly where our framework shows an advantage: even under the more challenging multi-environment joint finetuning setup, our method still surpasses conventional baselines that are trained under the easier single-benchmark setting, while also requiring fewer finetuning steps. We will clarify these setups and their fairness implications more explicitly in the revision.
>
> **Reference**
> [1] Li L, Zhang Q, Luo Y, et al. Causal World Modeling for Robot Control[J]. arXiv preprint arXiv:2601.21998, 2026.
> [2] Bi H, Tan H, Xie S, et al. Motus: A unified latent action world model[J]. arXiv preprint arXiv:2512.13030, 2025.

---

> > ### Author Rebuttal · Reviewer_motz · 2026-04-02
> >
> > The author provides the direct results or relevant evidence for each question. Among them, the VLM finetuning setup is clarified in a clear way, while the comparison with other VLAs/WAs and the justification for long-horizon tasks are a little vague. Overall, this is a paper with interesting ideas. Although some of the concerns cannot be fully resolved, I would keep the original acceptance rating. I recommend that the author consider the comments in the future release.

---

### Official Review · Reviewer_qeix · 2026-03-12

**Soundness:** 3
**Presentation:** 3
**Significance:** 3
**Originality:** 2
**Overall Recommendation:** 5
**Confidence:** 4

**Summary:**

This paper investigates a novel interface for adapting Vision-Language Models (VLMs) to robotic control by utilizing sparse 3D waypoints as the generative target. While these waypoints could theoretically be executed via manual inverse kinematics (IK), the authors instead propose a Generalizable Action Expert (GAE). This task-agnostic expert learns to translate sparse spatial guidance into precise, continuous action trajectories by conditioning on real-time point cloud observations. Extensive experiments across both simulation benchmarks and real-world environments demonstrate the effectiveness of this decoupled framework. The authors also provide comprehensive ablation studies to validate their core design choices.

**Compliance With Llm Reviewing Policy:**

Affirmed.

**Final Justification:**

The methodology is innovative and well-motivated, offering a promising path toward more adaptable training targets for robotics foundation models. While the initial submission contained some overstated claims and lacked certain discussions regarding related work, the authors' response has addressed these concerns effectively. One area for future exploration is a more granular analysis of frame extraction—specifically, investigating how different sparse 3D point trajectories adapt to diverse scenarios.

**Key Questions For Authors:**

Could the authors clarify the exact size of the SFT dataset after keyframe extraction? Does this 1k-step SFT budget actually iterate through the entirety of the multi-domain data, or is the VLM only trained on a heavily subsampled fraction?

**Limitations:**

Yes

**Strengths And Weaknesses:**

#### Strength
1. The framework effectively resolves a major bottleneck in current VLA models by cleanly separating high-level semantic reasoning from low-level geometric motor control. This sparse 3D interface successfully mitigates the catastrophic forgetting often seen when fine-tuning VLMs on dense, domain-specific continuous actions.
2. The experimental results are compelling, the super strong adaption and environment-agnostic feature of point cloud particularly the model's zero-shot generalization capabilities.
3. The authors provide thorough ablation studies that successfully validate their core design choices. The paper systematically breaks down the impact of VLM fine-tuning steps, depth noise robustness, interpolation methods, and the APPF training strategy, offering valuable empirical insights into the system's mechanics.


#### Weakness
1. The claim that "In all simulation experiments, the action expert is deployed in a zero-shot manner(frozen)"  is misleading and technically incorrect. According to the dataset details, the pre-training data for the action expert includes trajectories from the evaluated simulation environments (RoboTwin and BridgeDataV2). While the action expert might be frozen during the VLM's supervised fine-tuning phase, it has already been trained on the simulation data distribution.
2. The paper lacks discussion and comparison to critical baseline methods in the realm of VLM-to-robot interfaces, most notably VoxPoser (Huang, Wenlong, et al., "VoxPoser: Composable 3D Value Maps for Robotic Manipulation with Language Models"). VoxPoser and its follow-up works also propose prompting strong VLMs/LLMs to generate spatial affordance/value maps as a generalized interface. A discussion contrasting the proposed sparse 3D waypoint interface with dense 3D value maps is necessary.
3. The training of the VLM relies heavily on human-designed data curation heuristics that may limit scalability and generalization. For example, the selection of keyframes is explicitly based on changes in the gripper's kinematic state (e.g., opening or closing). While this heuristic is suitable for standard pick-and-place tasks, it could introduce significant flaws or fail to capture necessary intermediate motions for complex long-horizon tasks or dexterous manipulation.


#### Minor Weakness:
Typo in line 336: "Resuls" -> "Results"
Broken Reference in line 75

---

> ### Author Rebuttal · Authors · 2026-03-31
>
> We sincerely thank the reviewer for the positive assessment and constructive feedback. We address the main concerns below.
>
> > 💬 **W1 & Q1:** *Clarification of “zero-shot” vs. frozen deployment.*
>
> We thank the reviewer for pointing this out. We agree that our original wording overstated “zero-shot” deployment. For RoboTwin and BridgeDataV2, the accurate statement is that GAE is deployed frozen without task-specific finetuning, rather than strict zero-shot domain generalization. We will revise this wording throughout the paper.
>
> At the same time, we would like to clarify that we do provide a stricter zero-shot simulation result in the appendix: GAE is directly transferred to ManiSkill without finetuning, and ManiSkill is not used in GAE pretraining. In this setting, frozen GAE achieves **0.89 / 0.84 / 0.62** success on Push Cube / Stack Cube / Pull Cube, outperforming DP3 **(0.83 / 0.76 / 0.48)** and ACT **(0.81 / 0.69 / 0.40)**, both of which are trained on the target domain. We will make this result more explicit in the revision. More broadly, all of our real-world experiments also use the action expert in a zero-shot manner: GAE is not trained on any our real-world setting collected point cloud or trajectory data, and is transferred directly from pretraining to real-robot deployment without task-specific finetuning. In the revision, we will clearly separate the claim to avoid ambiguity.
>
> > 💬 **W2 & Q2:** *On comparison with VoxPoser-style interfaces.*
>
> We thank the reviewer for this important suggestion. VoxPoser and its follow-up works show that dense 3D value maps are a strong interface for language-conditioned spatial reasoning: LLMs/VLMs compose affordance and constraint maps in voxel space, and motion planning / MPC searches for trajectories on top of them. Thus, dense value maps are primarily a planning interface, rather than a directly executable interface to a learnable low-level expert.
>
> By contrast, our sparse 3D waypoint interface is designed to make the action expert itself general and scalable. The high-level VLM outputs sparse task-critical geometric guidance, and GAE refines it into executable motion from point clouds. This reduces ambiguity and semantic burden on the low-level controller, while enabling a reusable learnable action expert that can improve through large-scale interaction data. We also add a direct comparison below, where our method performs consistently better across short-, middle-, and long-horizon in-domain settings:
>
> | Method | Short Horizon | Middle Horizon | Long Horizon |
> |:---|:---:|:---:|:---:|
> | VoxPoser | 0.62 | 0.52 | 0.28 |
> | VLM+GAE (Ours) | **0.82** | **0.75** | **0.61** |
>
> > 💬 **W3 & Q3:** *On scalability of keyframe extraction heuristics.*
>
> We appreciate this concern and agree that the current paper should better clarify the role and limitation of keyframe extraction. Our intention is not to claim that the current heuristic is universally optimal, nor that sparse supervision must depend on gripper kinematics. Rather, the key idea is to construct a general sparse geometric supervision signal aligned with the proposed interface, and the current kinetic-keyframe rule is only one simple implementation.
>
> To directly address this point, we add a comparison between our current kinetic keyframe extraction (ours) and a more general uniform keyframe sampling strategy under matched supervision budgets. The results show that uniform sampling remains competitive, while kinetic keyframes are slightly better on most benchmarks. This suggests that the overall framework does not critically depend on hand-designed gripper-state rules, while the current heuristic still provides a useful inductive bias for the benchmarks considered here.
>
> **SimplerEnv**
>
> | Model | Google Robot | WidowX Grasp | WidowX Success |
> |:---:|:---:|:---:|:---:|
> | Uniform | 73.5% | 90.8% | 74.6% |
> | Kinetic (ours) | **74.1%** | **91.4%** | **75.1%** |
>
> **RoboTwin**
>
> | Model | Short Horizon | Middle Horizon | Long Horizon |
> |:---:|:---:|:---:|:---:|
> | Uniform | 0.81 | 0.74 | **0.61** |
> | Kinetic (ours) | **0.82** | **0.77** | 0.60 |
>
> > 💬 **Q4:** *On the exact SFT dataset size and what 1k steps cover.*
>
> Thank you for raising this important question. We sincerely apologize that Table 4 was not written clearly enough and may have caused misunderstanding. In “8×32×1k,” 32 is the total batch size across 8 GPUs, so the per-GPU batch size is 4. After keyframe extraction, our simulation SFT set contains 31,079 samples from SIMPLEREnv (BridgeData + Fractal) and RoboTwin, organized as standard image-grounded VLM QA samples. Thus, 1k steps ≈ 32,000 training instances, which is essentially one full pass over the extracted multi-domain SFT set. We will clarify this dataset size and coverage in the revision, and will open-source both the training data (for VLM and GAE) and the training code.

---

> > ### Author Rebuttal · Reviewer_qeix · 2026-04-01
> >
> > I thank the authors for their rebuttal, which addressed all my primary concerns. The methodology is innovative and well-motivated, offering a promising path toward more adaptable training targets for robotics foundation models. I have raised my score accordingly. One area for future exploration is a more granular analysis of frame extraction, specifically regarding how different sparse 3D point trajectories adapt to diverse scenarios.

---

### Official Review · Reviewer_RPts · 2026-03-13

**Soundness:** 4
**Presentation:** 3
**Significance:** 4
**Originality:** 4
**Overall Recommendation:** 5
**Confidence:** 3

**Summary:**

This paper presents GAE, a modular Vision-Language-Action framework that cleanly separates high-level semantic reasoning from low-level motor control through a sparse 3D pose interface. By introducing a task-agnostic Generalizable Action Expert trained on a large-scale pointcloud–trajectory dataset, the method significantly improves generalization while mitigating catastrophic forgetting in VLMs. The proposed APPF training paradigm further enhances efficiency and robustness. Extensive simulation and real-world experiments demonstrate strong performance across diverse tasks, viewpoints, and environments, particularly in long-horizon settings. Overall, the work offers a principled and scalable solution toward reusable and general robotic manipulation systems.

**Compliance With Llm Reviewing Policy:**

Affirmed.

**Final Justification:**

I maintain my original score and recommendation for acceptance.

**Key Questions For Authors:**

Weaknesses are minor and mainly related to presentation and discussion depth rather than technical soundness.

**Limitations:**

yes

**Strengths And Weaknesses:**

Strengths
- Clear architectural insight: Elegant decoupling of semantic reasoning and geometric control via a sparse 3D pose interface.
- Strong generalization: Demonstrates robustness across novel viewpoints, objects, environments, and long-horizon tasks.
- Mitigates catastrophic forgetting: Requires only lightweight VLM fine-tuning while preserving language reasoning ability.
- Well-designed training paradigm (APPF): Efficient two-stage training improves convergence and data utilization.
- Large-scale dataset construction: 150k pointcloud–trajectory pairs from both simulation and real-world sources.
- Comprehensive evaluation: Includes simulation, real-world, OOD, robustness, and ablation studies.
- Practical deployment advantages: Higher inference speed and robustness to depth noise.

Weakness
- Minor presentation issues (e.g., small typos and formatting inconsistencies in Section 2.2 references).
- Some comparisons with closely related modular VLA or hierarchical methods could be discussed more thoroughly.
- Limited theoretical analysis explaining why sparse waypoint supervision better preserves VLM capabilities.

---

> ### Author Rebuttal · Authors · 2026-03-30
>
> We thank the reviewer for the positive assessment and for recognizing the strengths of our method in design, generalization, and evaluation.
>
> Regarding the minor presentation issues, we appreciate the careful reading and will correct the typos and formatting inconsistencies in the camera-ready version.
>
> >💬 **W2 & Q2**: *Some comparisons with closely related modular VLA or hierarchical methods could be discussed more thoroughly.*
>
> We thank the reviewer for this helpful suggestion and will clarify this comparison.
>
> Our contribution is not modularization alone, but showing that the **action expert itself can be general and scalable**.  In many recent hierarchical VLAs, such as HAMSTER, BridgeVLA, ThinkAct, and GR00T N1, the emphasis remains on scaling the high-level VLM, while the low-level controller is still task-dependent, tightly coupled to upstream representations. As a result, they do not fully show that the action expert can be trained once and reused across tasks.
>
> In contrast, our goal is to build a **generalizable action expert (GAE)** that transfers across tasks without task-specific finetuning. The sparse 3D geometric interface is a means to this end: it removes most semantic burden from the low-level controller, allowing GAE to focus on geometry-aware motion refinement and enabling cross-task reuse.
>
> A second distinction is from planner-style modular methods like SoFar, PointWorld [1], and Dream2Flow [2]. While these methods provide clean interfaces, they typically rely on non-learnable planning, rather than a learnable action expert. As a result, they do not support a reusable action expert that can improve through large-scale interaction data. This difference is especially important for rapid local adaptation, contact-rich manipulation, and obstacle-aware motion refinement.
>
> >💬 **W3 & Q3**: *Limited theoretical analysis explaining why sparse waypoint supervision better preserves VLM capabilities.*
>
> We thank the reviewer for this helpful comment. We agree that the intuition behind our design could be stated more clearly. Our design is supported by several principled considerations.
>
> **First**, long robot trajectories are highly redundant: among hundreds of low-level actions, only a few key geometric decision points are task-critical, while many intermediate steps mainly reflect local servoing and smooth interpolation. Let $A =(a1,…,aT)$ denote the dense action trajectory and $W=(w1,…,wK)$, with $K≪T$ denote sparse task-critical waypoints. Dense and sparse supervision can be written as:
> $\mathcal{L}\_{\mathrm{dense}} = \sum\_{t=1}^{T} \ell\left(f\_{\theta}(o_t), a_t\right)$,
> $\mathcal{L}\_{\mathrm{sparse}} = \sum\_{k=1}^{K} \ell\left(f_{\theta}(o_{t_k}), w_k\right).$
> From an information-bottleneck perspective, sparse waypoint supervision can be viewed as a compressed target that preserves task-relevant information while discarding redundant embodiment-specific control details. In contrast, dense action supervision forces the VLM to model low-level control patterns at every step, which increases task-specific drift.
>
> **Second**, this also affects optimization. When temporally adjacent actions are highly correlated, dense supervision spreads gradient across many redundant, low-information steps, whereas sparse supervision concentrates learning signal on a small number of decision-critical points:
> $\nabla\_{\theta}\mathcal{L}\_{\mathrm{dense}} = \sum\_{t=1}^{T} g_t$,
> $\nabla\_{\theta}\mathcal{L}\_{\mathrm{sparse}} = \sum\_{k=1}^{K} g_{t_k}.$
> This encourages the VLM to focus on high-level geometric reasoning, while delegating embodiment-specific low-level refinement to GAE.
>
> **Third**, sparse waypoint prediction is substantially closer to the VLM pretraining distribution than dense action prediction. Our VLM predicts anchors/waypoints in an **image- or camera-aligned space**, which is more consistent with the image-grounded outputs VLMs are naturally pretrained on. Moreover, instead of introducing action tokenizers or manually discretized action bins, we represent actions directly in **text**, keeping the prediction space much closer to the semantic space of LLM/VLM pretraining. Recent works [3,4] similarly show that text-based action representations better preserve pretrained VLM capabilities while improving generalization.
>
>
> **References**
> [1] Huang W, Chao Y W, Mousavian A, et al. PointWorld: Scaling 3D World Models for In-The-Wild Robotic Manipulation[J]. arXiv preprint arXiv:2601.03782, 2026.
> [2] Dharmarajan K, Huang W, Wu J, et al. Dream2Flow: Bridging Video Generation and Open-World Manipulation with 3D Object Flow[J]. arXiv preprint arXiv:2512.24766, 2025.
> [3] Zha L, Hancock A J, Zhang M, et al. LAP: Language-Action Pre-Training Enables Zero-shot Cross-Embodiment Transfer[J]. arXiv preprint arXiv:2602.10556, 2026.
> [4] Hancock A J, Wu X, Zha L, et al. Actions as language: Fine-tuning vlms into vlas without catastrophic forgetting[J]. arXiv preprint arXiv:2509.22195, 2025.

---

> > ### Author Rebuttal · Reviewer_RPts · 2026-04-01
> >
> > The authors have adequately addressed all of my concerns in the rebuttal. I appreciate the clarified contributions and additional explanations provided.
> >
> > I find the novelty of this work and the amount of effort behind it to be both significant and commendable. The proposed approach demonstrates strong performance, especially in the context of generally limited generalization ability observed in many existing models, which makes this contribution particularly valuable to the field.
> >
> > I also reviewed the concerns raised by other reviewers and the corresponding author responses. Overall, I do not find any remaining issues that would affect the validity or impact of the paper.
> >
> > I strongly support acceptance. Additionally, I encourage the authors to consider open-sourcing their code and resources, which would further benefit and accelerate progress in the research community.

---

### Decision · Program_Chairs · 2026-04-30

**Decision:**

Accept (regular)

**Comment:**

The paper introduces GAE, a modular VLA framework that leverages a VLA for high-level planning to predict sparse 3D waypoints, combined with a low-level motion controller based on a 3D diffusion policy. Both simulation and real-world robot experiments demonstrate the effectiveness of the proposed approach.

The paper was reviewed by four reviewers. The main concerns raised include:
- Presentation issues (RPts, motz, 4rNU), with suggestions to improve the clarity and quality of writing.
- Insufficient discussion and comparison with related modular or state-of-the-art VLA methods (RPts, qeix, motz, 4rNU).
- Limited analysis of why sparse waypoints are advantageous (RPts).
- Misleading or overstated claims (qeix, 4rNU).
- Heuristic keyframe selection strategy (qeix).

The rebuttal addressed these concerns well, and all four reviewers unanimously recommend acceptance. The AC also supports acceptance.

For the final version, the authors should revise the paper according to reviewers' comments:
- Improve the overall presentation and clarity of writing.
- Revise claims to avoid overstatement.
- Expand discussion or comparison with relevant 3D manipulation methods, such as VoxPoser (Huang et al., 2023) and Gondola (Chen et al., 2025).
- Provide additional analysis on the effectiveness of sparse waypoints.

The authors are also encouraged to open-source their code, models, and datasets to enhance reproducibility and impact.